# *LongLLMLingua*: Accelerating and Enhancing LLMs in Long Context Scenarios via Prompt Compression

## Abstract

In long context scenarios, large language models (LLMs) face three main challenges: higher computational/financial cost, longer latency, and inferior performance. Some studies reveal that the performance of LLMs depends on both the density and the position of the key information (question relevant) in the input prompt. Inspired by these findings, we propose LongLLMLingua for prompt compression towards improving LLMs' perception of the key information to simultaneously address the three challenges. We conduct evaluation on a wide range of long context scenarios including single-/multi-document QA, few-shot learning, summarization, synthetic tasks, and code completion. Experimental results show that LongLLMLingua compressed prompt can derive higher performance with much lower cost. The latency of the end-to-end system is also reduced. For example, on NaturalQuestions benchmark, LongLLMLingua gains a performance boost of up to 17.1% over the original prompt with ∼4x fewer tokens as input to GPT-3.5-Turbo. It can drive cost savings of $28.5 and $27.4 per 1,000 samples from the LongBench and ZeroScrolls benchmark, respectively. Additionally, when compressing prompts of ∼10k tokens at a compression rate of 2x-10x, LongLLMLingua can speed up the end-to-end latency by 1.4x-3.8x.

## 1 Introduction

ChatGPT and other large language models (LLMs) have revolutionized user-oriented language technologies and are serving as crucial components in more and more applications. Carefully designing prompts is necessary to achieve better performance in specific downstream tasks. The commonly used technologies such as In-Context Learning (ICL) (Dong et al., 2023), Retrieval Augment Generation (RAG) (Lewis et al., 2020), and Agent (Park et al., 2023) are driving prompts to be increasingly longer, even reaching thousands of tokens. Scenarios such as multi-document question answering, code completion, and document summarization also necessitate the processing of long contexts.

There are three main challenges when LLMs are used in long context scenarios: (1) The higher computational and financial cost required to run these models or to call APIs from companies providing LLM services. This can be a significant barrier for individuals or smaller organizations with limited resources. (2) The longer latency associated with LLMs, which can cause delays in generating responses or predictions and is particularly problematic in real-time scenarios where users expect quick and accurate responses. (3) The inferior performance caused by the extended window size of LLMs (Xiong et al., 2023), and the low density as well as the less sensitive position of the question-relevant key information in the prompt. Figure 1a shows that LLMs' performance in downstream tasks may decrease as the noisy information in the prompt increases (Shi et al., 2023). Moreover, the purple curve in Figure 1b indicates that LLMs' ability to capture the relevant information depends on their positions in the prompt (Liu et al., 2023): they achieve the highest performance when relevant information occurs at the beginning or end of the input context, and significantly degrades if relevant information is located in the middle of long contexts.

Inspired by these observations, we propose *LongLLMLingua* to address the three challenges. Specifically, we use the advanced while efficient LLMLingua (Jiang et al., 2023a) as our backbone framework for prompt compression to address the first two challenges, *i.e.*, reduce cost and latency. How-

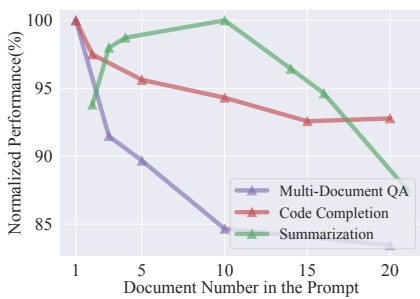 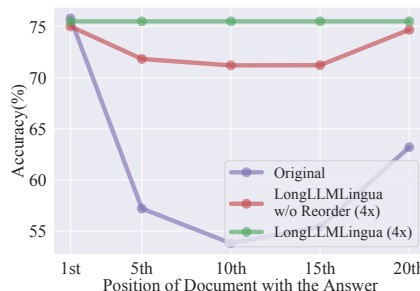

(a) Performance v.s. Document Number      (b) Performance v.s. Key Information Position

Figure 1: (a) LLMs' performance in downstream tasks may decrease as the noisy information in the prompt increases. In this case, we keep $k$ most relevant documents/paragraphs based on the ground truth or LongLLMLingua $r_k$. A larger $k$ implies more noise introduced into the prompt. To improve the key information density in the prompt, we present question-aware coarse-to-fine compression. (b) LLMs' ability to capture the relevant information depends on their positions in the prompt. To reduce information loss in the middle, we introduce a document reordering mechanism.

ever, in the case of long contexts, the distribution of question-relevant key information in the prompt is generally sparse. Existing prompt compression methods like LLMLingua (Jiang et al., 2023a) and Selective-Context (Li, 2023) that do not consider the content of the question during compression may retain too much noisy information in the compressed results, leading to inferior performance. In this paper, LongLLMLingua is designed to enhance LLM's perception of key information (relevant to the question) in the prompt, so that the third challenge of inferior performance in long context scenarios could be addressed. Figure 1b is an example. The underlying principle of LongLLMLingua is that small language models are inherently capable of capturing the distribution of key information relevant to a given question.

Our main contributions are five-fold: (1) We propose a question-aware coarse-to-fine compression method to improve the key information density in the prompt (Sec. 4.1); (2) We introduce a document reordering mechanism to reduce information loss in the middle. (Sec. 4.2); (3) We present dynamic compression ratios to bridge the coarse-grained compression and fine-grained compression for adaptive granular control (Sec. 4.3); (4) We propose a post-compression subsequence recovery strategy to improve the integrity of the key information (4.4). (5) We evaluate LongLLMLingua on three benchmarks, *i.e.*, NaturalQuestions (Liu et al., 2023), LongBench (Bai et al., 2023), and ZeroSCROLLS (Shaham et al., 2023). Experimental results demonstrate that compared with original prompts, LongLLMLingua compressed prompts can achieve higher performance with much lower costs. The latency of the end-to-end system is also reduced.

## 2 PROBLEM FORMULATION

Following LLMLingua (Jiang et al., 2023a), we use $\mathbf{x} = (\mathbf{x}^{\text{ins}}, \mathbf{x}_1^{\text{doc}}, \cdots, \mathbf{x}_K^{\text{doc}}, \mathbf{x}^{\text{que}})$ to represent a prompt, which composed of the instruction $\mathbf{x}^{\text{ins}}$, $K$ documents $\mathbf{x}_i^{\text{doc}}$, and the question $\mathbf{x}^{\text{que}}$. In fact, the prompt can be modified according to specific application scenarios. For example, $\mathbf{x}^{\text{ins}}$ at the beginning can be removed, $\mathbf{x}^{\text{que}}$ can be any requirement specified by users, and $(\mathbf{x}_1^{\text{doc}}, \cdots, \mathbf{x}_K^{\text{doc}})$ can be any additional materials that users append to the prompt to get a better response from LLMs for $\mathbf{x}^{\text{que}}$. The objective of a prompt compression system can be formulated as:

$$\min_{\widetilde{\mathbf{x}}} D\left(\mathbf{y}, \widetilde{\mathbf{y}}\right) + \lambda \|\widetilde{\mathbf{x}}\|_0, \tag{1}$$

where $\widetilde{\mathbf{x}}$ denotes the compressed prompt and is a token-level subsequence of $\mathbf{x}$. $\mathbf{y}$ represents the ground-truth output texts with $\mathbf{x}$ as the input and $\widetilde{\mathbf{y}}$ represent the LLM-generated results derived by $\widetilde{\mathbf{x}}$. $D$ is a distance measure between two distributions, such as KL divergence. We expect the distribution of $\mathbf{y}$ and $\widetilde{\mathbf{y}}$ to be as similar as possible. $\lambda$ is a trade-off hyper-parameter regarding the compression ratio. In this work, we additionally incorporate an operation space of permutation over the $K$ documents $(\mathbf{x}_1^{\text{doc}}, \cdots, \mathbf{x}_K^{\text{doc}})$ for joint optimization.

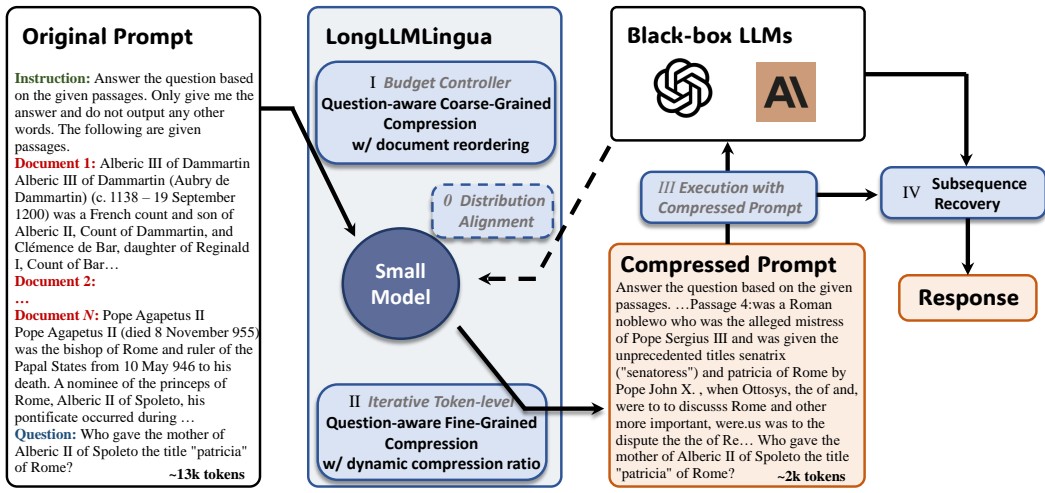

Figure 2: Framework of *LongLLMLingua*. Gray *Italic* content: As in LLMLingua.

# 3    PRELIMINARY: LLMLINGUA

LLMLingua (Jiang et al., 2023a) uses a small language model $\mathcal{M}_S$ to calculate the perplexity of each token in the original prompt and then removes tokens with lower perplexities. The rationale behind this approach is that tokens with lower perplexities contribute less to the overall entropy gain of the language model, so removing them will have a relatively minor impact on the LLM's comprehension of the context. LLMLiungua consists of three components: a budget controller, an iterative token-level prompt compression algorithm, and a distribution alignment mechanism, as shown by Italic texts in Figure 2. The budget controller allocates different compression ratios to the various components in the original prompt (*i.e.*, instruction, demonstrations, question), and performs coarse-grained compression at the demonstration level. The intermediate results are divided into segments and the token-level compression is then performed segment by segment, with the perplexity of each token conditioned on previous compressed segments calculated by $\mathcal{M}_S$. For distribution alignment, it performs instruction tuning on $\mathcal{M}_S$ with the data generated by the target LLM to narrow the gap between the distribution of LLM and that of $\mathcal{M}_S$ used for prompt compression.

# 4    LONGLLMLINGUA

LongLLMLingua is developed upon the framework of LLMLingua towards prompt compression in long context scenarios. The primary challenge in long context scenarios is how to enhance LLM's perception of key information relevant to the question in the prompt. LongLLMLingua addresses this challenge from three perspectives, and further applies a subsequence recovery strategy to improve the accuracy and reliability of the information provided to users. We elaborate on each component in this section.

## 4.1    HOW TO IMPROVE KEY INFORMATION DENSITY IN THE PROMPT?

**Question-Aware Coarse-Grained Compression**    In coarse-grained compression, we aim to figure out a metric $r_k$ to evaluate the importance of each document $\mathbf{x}_k^{\text{doc}} = \{x_{k,i}^{\text{doc}}\}_{i=1}^{N_k}$, where $N_k$ is the number of tokens in $\mathbf{x}_k^{\text{doc}}$. We only keep $\mathbf{x}_k^{\text{doc}}$ with higher $r_k$ as the intermediate compressed results.

LLMLingua uses document-level perplexity to represent the importance of documents: $r_k = 1/N_k \sum_i^{N_k} p(x_{k,i}^{\text{doc}}) \log p(x_{k,i}^{\text{doc}}), k \in \{1, 2, \cdots, K\}$. Although the retained documents typically contain a lot of information, they are irrelevant to the question $\mathbf{x}^{\text{que}}$ and instead become noise, reducing key information density in the compressed results and bringing difficulties for LLM to output correct answers. As shown in Figure 3a, the recall@16 of LLMLingua only reaches 50%, indicating its incompetence in retaining key information during compression.

Retrieval-based methods are also feasible here. We can use $\mathbf{x}^{\text{que}}$ to retrieve the most relevant documents among $(\mathbf{x}_1^{\text{doc}}, \cdots, \mathbf{x}_K^{\text{doc}})$ as the compressed results. However, these methods struggle to distinguish question-related fine-grained semantic information. Some documents with key information may be discarded during retrieval. As shown in Figure 3a, embedding-based methods such as Sentence BERT and OpenAI Embedding only achieve ∼75% accuracy in recall@5, which implies that the final accuracy upper bound of LLMs with 4x compression is only 75%.

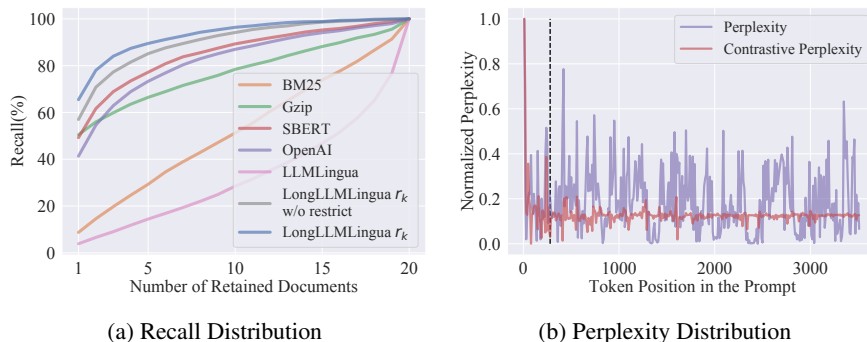

(a) Recall Distribution  (b) Perplexity Distribution

Figure 3: (a) Comparison of recall on NaturalQuestions Multi-documemnt QA dataset. (b) Comparison between perplexities and contrastive perplexities of tokens in the prompt from Multi-documemnt QA dataset. The document with the ground truth is located on the left side of the dashed line.

One approach to improve key information density in the compressed results is to calculate document-level perplexity conditioned on the question $\mathbf{x}^{\text{que}}$. However, this method may not be effective because documents often contain a significant amount of irrelevant information. Even when conditioned on $\mathbf{x}^{\text{que}}$, the perplexity scores computed for entire documents may not be sufficiently distinct, rendering them an inadequate metric for document-level compression. Therefore, we propose to use the perplexity of the question $\mathbf{x}^{\text{que}}$ conditioned on different contexts $\mathbf{x}_k^{\text{doc}}$ to represent the association between them. We append a restrictive statement $\mathbf{x}^{\text{restrict}}$[1] after $\mathbf{x}^{\text{que}}$ to strengthen the interconnection of $\mathbf{x}^{\text{que}}$ and $\mathbf{x}_k^{\text{doc}}$. It can be regarded as a regularization term that mitigates the impact of hallucinations. This can be formulated as:

$$r_k = \frac{1}{N_c} \sum_i^{N_c} p(x_i^{\text{que,restrict}}|\mathbf{x}_k^{\text{doc}}) \log p(x_i^{\text{que,restrict}}|\mathbf{x}_k^{\text{doc}}), k \in \{1, 2, \cdots, K\}, \qquad (2)$$

where $x_i^{\text{que,restrict}}$ is the $i$-th token in the concatenated sequence of $\mathbf{x}^{\text{que}}$ and $\mathbf{x}^{\text{restrict}}$ and $N_c$ in the number of tokens.

Figure 3a demonstrates that our coarse-level compression approach achieves the highest recall with different numbers of retained documents, suggesting that it preserves the most key information from the documents $(\mathbf{x}_1^{\text{doc}}, \cdots, \mathbf{x}_K^{\text{doc}})$ in the compressed results.

**Question-Aware Fine-Grained Compression**  In fine-grained compression, we assess the importance of each token in the instruction $\mathbf{x}^{\text{ins}}$, the question $\mathbf{x}^{\text{que}}$, and $K'$ documents $\{\mathbf{x}_i^{\text{doc}}\}_{i=1}^{K'}$ retained after coarse-grained compression. We incorporate the iterative compression mechanism following LLMLingua and directly calculate token perplexities to compress $\mathbf{x}^{\text{ins}}$ and $\mathbf{x}^{\text{que}}$. In this section, we investigate how to make the fine-grained token-level compression over $\{\mathbf{x}_k^{\text{doc}}\}_{k=1}^{K'}$ aware of the question $\mathbf{x}^{\text{que}}$, so that the compressed results could contain more question-relevant key information.

A straightforward solution for the awareness of $\mathbf{x}^{\text{que}}$ is to simply concatenate it at the beginning of the whole context. However, this will result in low perplexities of relevant tokens in the context following the condition, further reducing their differentiation from general tokens. In this paper, we propose *contrastive perplexity*, *i.e.*, the distribution shift caused by the condition of the question, to represent the association between the token and the question. The contrastive perplexity based

---

[1]Specifically, "*We can get the answer to this question in the given documents*".

importance metric $s_i$ for each token $x_i$ in $\{\mathbf{x}_k^{\text{doc}}\}_{k=1}^{K'}$ can be formulated as:

$$s_i = \text{perplexity}(x_i|x_{<i}) - \text{perplexity}(x_i|x^{\text{que}}, x_{<i}). \tag{3}$$

Figure 3b illustrates the difference between perplexities and contrastive perplexities. We can see that tokens of high perplexities are widely distributed in all documents. However, tokens with high contrastive perplexities concentrate more on the left side of the dashed line, which corresponds to the document that contains the answer to the question. This suggests that the proposed contrastive perplexity can better distinguish tokens relevant to the question, thus improving the key information density in the compressed results.

## 4.2 HOW TO REDUCE INFORMATION LOSS IN THE MIDDLE?

As demonstrated in Figure 1b, LLM achieves the highest performance when relevant information occurs at the beginning and significantly degrades if relevant information is located in the middle of long contexts. After the coarse-grained compression, we have obtained a set of documents $\{\mathbf{x}_k^{\text{doc}}\}_{k=1}^{K'}$ with their corresponding importance scores $\{r_k\}_{k=1}^{K'}$ indicating their association with the question $\mathbf{x}^{\text{que}}$. Therefore, we reorder documents using their importance scores to better leverage LLMs' information perception difference in positions:

$$(\mathbf{x}^{\text{ins}}, \mathbf{x}_1^{\text{doc}}, \cdots, \mathbf{x}_{K'}^{\text{doc}}, \mathbf{x}^{\text{que}}) \xrightarrow{r_k} (\mathbf{x}^{\text{ins}}, \mathbf{x}_{r1}^{\text{doc}}, \cdots, \mathbf{x}_{rK'}^{\text{doc}}, \mathbf{x}^{\text{que}}) \tag{4}$$

## 4.3 HOW TO ACHIEVE ADAPTIVE GRANULAR CONTROL DURING COMPRESSION?

In fine-grained compression, LLMLingua applies the save compression ratio over all documents obtained from coarse-grained compression. However, the key information density of different documents is different. The more relevant to the question a document is, the more budget (*i.e.*, lower compression ratio) we should allocate to it. Therefore, we bridge coarse-grained compression to fine-grained compression and use the importance scores $\{r_k\}_{k=1}^{K'}$ obtained from coarse-grained compression to guide the budget allocation in fine-grained compression. In this way, we can achieve adaptive granular control on the whole.

Specifically, we first determine the initial budget for the retained documents $\tau^{\text{doc}}$ [2] using the budget controller of LLMLingua. During fine-grained compression, we follow the iterative token-level compression algorithm in LLMLingua but dynamically assign the compression budget $\tau_k^{\text{doc}}$ to each document $\mathbf{x}_k^{\text{doc}}$ according to the ranking index $I(r_k)$ (e.g., 0, 1) of its importance score from the coarse-grained compression. In this paper, we employ a linear scheduler for the adaptive allocation. Budget of each token $x_i$ can be formulated as:

$$\tau_i = \tau_k^{\text{doc}}, \quad \forall x_i \in \mathbf{x}_k^{\text{doc}},$$
$$\tau_k^{\text{doc}} = \max(\min((1 - \frac{2I(r_k)}{K'})\delta\tau + \tau^{\text{doc}}, 0), 1), \tag{5}$$

where $i$ and $k$ is the index of token and document, $K'$ denotes the number of documents, and $\delta\tau$ is a hyper-parameter that controls the overall budget for dynamic allocation.

## 4.4 HOW TO IMPROVE THE INTEGRITY OF KEY INFORMATION?

Certain tokens of key entities may be discarded during the fine-grained token-wise compression. For example, the time entity "2009" in the original prompt might be compressed to "209" and the name entity "Wilhelm Conrad Röntgen" might be compressed to "Wilhelmgen". This can cause problems for fact-based tasks like document QA, where language models tend to replicate information from the prompt, as shown in Figure 4.

To improve the accuracy and reliability of the information provided to users, we propose a subsequence recovery method to restore the original content from LLMs' responses. This method relies on the subsequence relationship among tokens in the original prompt, compressed prompt, and LLMs'

---

[2] In LLMLingua, it is $\tau^{\text{dems}}$ for demonstrations.

| Document [1](Title: List of Nobel laureates in Physics) The first Nobel Prize in Physics was awarded in 1901 to {Wilhelm Conrad Röntgen}{Wilhelm Con rad Rö nt gen}, of Germany,...

**Original Prompt** | Document [1](Title: List of Nobelates in Physics) The first Nobel1 {Wilhelmgen}{Wilhelm gen}, of, who received, ....

**Compressed Prompt** | {Wilhelmgen}
{Wilhelm gen}

**LLMs' Response** |

Figure 4: The example of Subsequence Recovery, the red text represents the original text, and the blue text is the result after using the LLaMA 2-7B tokenizer.

response. The overall procedure includes: i) Iterate through tokens $y_l$ in LLMs' response and select the longest substring $\widetilde{y}_{\text{key},l} = \{y_l, y_{l+1}, ..., y_r\}$ that appears in the compressed prompt $\widetilde{x}$. ii) Find the maximum common shortest subsequence $x_{i,j} = \{x_i, x_{i+1}, ..., x_j\}$ in the original prompt $x$, corresponding to the representation $\widetilde{y}_{\text{key},l}$ in the original prompt (accelerated using prefix trees or sequence automata). iii) Replace the matched tokens $\widetilde{y}_{\text{key},l}$ in LLMs' response with the corresponding subsequence $x_{i,j}$ from the original prompt. For more details, please refer to Algorithm 1.

## 5 EXPERIMENTS

Here, we investigate: (1) How *effective* is LongLLMLingua? (2) How *efficient* is LongLLMLingua?

**Implementation details**  In this paper, we use GPT-3.5-Turbo-0613[3] and LongChat-13B-16k as the target LLMs, both accessible via OpenAI[4] and HuggingFace[5]. To ensure stable and reproducible results, we employ greedy decoding and set the temperature to 0 in all experiments. For the small language models used for compression, we apply LLaMA-2-7B-Chat[6], which has been aligned by supervised fine-tuning and RLHF. We implement our approach with PyTorch 1.13.1 and Hugging-Face Transformers. We set up hyperparameters following LLMLingua except for the segment size used in iterative token-level compression set to 200 here. More details are provided in Appendix C.

**Dataset & evaluation metric**  We use NaturalQuestions for the multi-document QA task, and use LongBench and ZeroSCROLLS for general long context scenarios. We also test on multi-hop QA tasks using MuSiQue dataset (Trivedi et al., 2022), and long dependency QA tasks using LooGLE benchmark (Li et al., 2023b). Please refer to Appendix D for more details on datasets.

(i) NaturalQuestions (Liu et al., 2023): This benchmark is similar to the retrieval-augmented generation setup in commercial search and question-answering scenarios like Bing Chat. Specifically, each question has 20 related documents in the original prompt. One of them contains the correct answer and there are five different ground truth document position settings in the prompt: 1st, 5th, 10th, 15th, and 20th. Following Liu et al. (2023), we use accuracy as the evaluation metric.

(ii) LongBench (Bai et al., 2023): This benchmark consists of six task types: single-document QA, multi-document QA, summarization, few-shot learning, code completion, and synthetic tasks. We used the English portion that covers 16 datasets for evaluation. We use the metrics and scripts provided along with the benchmark for evaluation.

(iii) ZeroSCROLLS (Shaham et al., 2023): This benchmark consists of four task types: summarization, QA, sentiment classification, and reordering, covering 10 datasets. We used the validation set for evaluation. We use the provided metrics and scripts for evaluation.

**Baselines**  We include two sets of baselines in following experiments:

*(i) Retrieval-based Methods.* We measure the association between the question and the documents in the prompt using five SoTA retrieval methods: BM25, Gzip (Jiang et al., 2023b), Sentence-BERT (Reimers & Gurevych, 2019), OpenAI Embedding, and the important metric $r_k$ used in LongLLMLingua coarse-grained compression. We discard sentences or paragraphs with low association until the compression constraint is met while keeping the original document order unchanged.

---

[3]For experiments with original prompts exceeding 4k tokens, we utilize GPT-3.5-Turbo-16k-0613.

[4]https://platform.openai.com

[5]https://huggingface.co/lmsys/longchat-13b-16k

[6]https://ai.meta.com/llama/

Table 1: Performance of different methods with different compression ratios on NaturalQuestions (20 documents) (Liu et al., 2023). Reorder: we reorder the documents with relevance metrics of different baselines as our document reordering strategy described in Sec. 4.2. In the case of OpenAI, it corresponds to LongContextReorder in the LangChain framework (Chase, 2022). For results reported under 1st to 20th, we do not use the reordering strategy for all methods.

| Methods | GPT3.5-Turbo | | | | | | LongChat-13b | | | | | | Length | | Latency | |
|---|---|---|---|---|---|---|---|---|---|---|---|---|---|---|---|---|
| | 1st | 5th | 10th | 15th | 20th | Reorder | 1st | 5th | 10th | 15th | 20th | Reorder | Tokens | $1/\tau$ | Latency | Speedup |
| *2x constraint* | | | | | | | | | | | | | | | | |
| *Retrieval-based Methods* | | | | | | | | | | | | | | | | |
| BM25 | 53.7 | 49.3 | 47.9 | 49.9 | 46.9 | 50.3 | 50.9 | 44.9 | 44.1 | 42.9 | 43.2 | 46.0 | 1,545 | 1.9x | 2.1 | 1.9x |
| Gzip | 64.6 | 63.8 | 60.5 | 58.3 | 57.3 | 64.4 | 61.9 | 55.7 | 52.7 | 50.8 | 50.9 | 59.3 | 1,567 | 1.9x | 2.1 | 1.9x |
| SBERT | 72.5 | 67.9 | 63.3 | 65.0 | 66.2 | 68.7 | 65.8 | 57.5 | 54.9 | 53.4 | 55.7 | 61.4 | 1,549 | 1.9x | 2.2 | 1.9x |
| OpenAI | 73.0 | 65.6 | 66.5 | 65.4 | 65.5 | 69.9 | 65.9 | 57.5 | 56.2 | 54.2 | 55.7 | 61.7 | 1,550 | 1.9x | 4.9 | 0.8x |
| LongLLMLingua $r_k$ | 73.9 | 67.7 | 68.7 | 66.0 | 65.6 | 74.3 | 68.5 | 59.1 | 56.8 | 55.3 | 56.9 | 65.2 | 1,548 | 1.9x | 2.3 | 1.8x |
| *Compression-based Methods* | | | | | | | | | | | | | | | | |
| Selective-Context | 45.4 | 39.0 | 33.8 | 33.5 | 41.5 | - | 53.2 | 26.3 | 25.4 | 24.2 | 33.3 | - | 1,478 | 2.0x | 7.4 | 0.6x |
| LLMLingua | 39.7 | 39.5 | 40.4 | 37.1 | 42.3 | 41.5 | 38.7 | 37.3 | 35.7 | 34.1 | 37.5 | 37.1 | 1,410 | 2.1x | 2.8 | 1.5x |
| **LongLLMLingua** | **77.2** | **72.9** | **70.8** | **70.5** | **70.6** | **76.2** | **68.7** | **59.4** | **57.3** | **55.9** | **58.4** | **66.1** | 1,429 | 2.1x | 2.9 | 1.4x |
| *4x constraint* | | | | | | | | | | | | | | | | |
| *Retrieval-based Methods* | | | | | | | | | | | | | | | | |
| BM25 | 40.6 | 38.6 | 38.2 | 37.4 | 36.6 | 36.3 | 39.5 | 37.5 | 36.8 | 36.4 | 35.5 | 37.7 | 798 | 3.7x | 1.5 | 2.7x |
| Gzip | 63.1 | 61.0 | 59.8 | 61.1 | 60.1 | 62.3 | 57.6 | 52.9 | 51.0 | 50.1 | 50.4 | 57.2 | 824 | 3.6x | 1.5 | 2.7x |
| SBERT | 66.9 | 61.1 | 59.0 | 61.2 | 60.3 | 64.4 | 62.6 | 56.6 | 55.1 | 53.9 | 55.0 | 59.1 | 808 | 3.6x | 1.6 | 2.5x |
| OpenAI | 63.8 | 64.6 | 65.4 | 64.1 | 63.7 | 63.7 | 61.2 | 56.0 | 55.1 | 54.4 | 55.0 | 58.8 | 804 | 3.7x | 4.3 | 1.0x |
| LongLLMLingua $r_k$ | 71.1 | 70.7 | 69.3 | 68.7 | 68.5 | 71.5 | 67.8 | 59.4 | 57.7 | 57.7 | 58.6 | 64.0 | 807 | 3.7x | 1.7 | 2.4x |
| *Compression-based Methods* | | | | | | | | | | | | | | | | |
| Selective-Context | 31.4 | 19.5 | 24.7 | 24.1 | 43.8 | - | 38.2 | 17.2 | 15.9 | 16.0 | 27.3 | - | 791 | 3.7x | 6.8 | 0.6x |
| LLMLingua | 25.5 | 27.5 | 23.5 | 26.5 | 30.0 | 27.0 | 32.1 | 30.8 | 29.9 | 28.9 | 32.4 | 30.5 | 775 | 3.8x | 1.8 | 2.2x |
| **LongLLMLingua** | **75.0** | **71.8** | **71.2** | **71.2** | **74.7** | **75.5** | **68.7** | **60.5** | **59.3** | **58.3** | **61.3** | **66.7** | 748 | 3.9x | 2.1 | 2.0x |
| Original Prompt | 75.7 | 57.3 | 54.1 | 55.4 | 63.1 | - | 68.6 | 57.4 | 55.3 | 52.5 | 55.0 | - | 2,946 | - | 4.1 | - |
| Zero-shot | | | 56.1 | | | | | | 35.0 | | | | 15 | 196x | 1.1 | 3.7x |

*(ii) Compression-based Methods.* We compare our approach with two state-of-art methods for prompt compression, *i.e.*, Selective Context (Li, 2023) and LLMLingua (Jiang et al., 2023a). Both methods employ LLaMA-2-7B-Chat as the small language model for compression. In LLMLingua, a coarse-to-fine approach is used to handle constraints of compression ratio: the original prompt is first compressed to $k$ times the constraint at a coarse level, where $k$ is the granular control coefficient; token-level is then performed to reach the overall constraint. Our method follows the same coarse-to-fine logic to achieve the constraint.

**Main results** Table 1 and 2 present the performance of various methods under different compression constraints. There are multiple observations and conclusions: (1) Our LongLLMLingua achieves the best performance across different tasks and constraints of compression ratios. Compared to the original prompt, our compressed prompt can derive higher performance with much lower cost. For example, LongLLMLingua gains a performance boost of 17.1% on NaturalQuestions with the ground-truth document at the 10th position, while the number of tokens input to GPT3.5-Turbo is ∼4x less. (2) Compression-based methods like Selective Context (Li, 2023) and LLMLingua (Jiang et al., 2023a) perform poorly on most tasks, especially those with abundant irrelevant information in the original prompt. This is due to their pure information entropy based compression mechanism, which includes too much noise in the compressed results and even leads to performance worse than the zero-shot setting, *e.g.*, on NaturalQuestions. (3) Retrieval-based methods work well with low compression rates. However, their performance declines as the compression progresses, *e.g.*, $2x \rightarrow 4x$; 3000 tokens → 2000 tokens. This may be caused by the decreased recall. Figure 3a is the illustration of cases on NaturalQuestions. (4) LongLLMLingua as well as our coarse-grained compression metric $r_k$ only is much more robust than all other baselines under different tasks and compression constraints. With the increase of the compression rate, *e.g.*, $2x \rightarrow 4x$, LongLLMLingua even achieves a little performance gain. We mainly owe this to the question-aware coarse-to-fine compression, which can better figure out the key information and reach a higher key

---

[6]https://python.langchain.com/docs/modules/data_connection/document_transformers/post_retrieval/long_context_reorder

Table 2: Performance of different methods under different compression ratios on LongBench (Bai et al., 2023) and ZeroSCROLLS (Shaham et al., 2023) using GPT-3.5-Turbo. Considering the dataset structure, we do not use the reordering strategy here.

| Methods | LongBench | | | | | | | | | | ZeroSCROLLS | | | |
|---|---|---|---|---|---|---|---|---|---|---|---|---|---|---|
| | SingleDoc | MultiDoc | Summ. | FewShot | Synth. | Code | AVG | Tokens | $1/\tau$ | Latency | AVG | Tokens | $1/\tau$ | Latency |
| *3,000 tokens constraint* | | | | | | | | | | | | | | |
| *Retrieval-based Methods* | | | | | | | | | | | | | | |
| BM25 | 32.3 | 34.3 | 25.3 | 57.9 | 45.1 | 48.9 | 40.6 | 3,417 | 3x | 7.5(2.1x) | 19.8 | 3,379 | 3x | 5.5(2.2x) |
| SBERT | 35.3 | 37.4 | 26.7 | 63.4 | 51.0 | 34.5 | 41.4 | 3,399 | 3x | 7.7(2.0x) | 24.0 | 3,340 | 3x | 5.9(2.1x) |
| OpenAI | 34.5 | 38.6 | 26.8 | 63.4 | 49.6 | 37.6 | 41.7 | 3,421 | 3x | 13.3(1.2x) | 22.4 | 3,362 | 3x | 11.7(1.0x) |
| LongLLMLingua $r_k$ | 37.6 | 42.9 | 26.9 | 68.2 | 49.9 | 53.4 | 46.5 | 3,424 | 3x | 8.2(1.9x) | 29.3 | 3,350 | 3x | 6.2(2.0x) |
| *Compression-based Methods* | | | | | | | | | | | | | | |
| Selective-Context | 23.3 | 39.2 | 25.0 | 23.8 | 27.5 | 53.1 | 32.0 | 3,328 | 3x | 50.6(0.3x) | 20.7 | 3,460 | 3x | 54.2(0.2x) |
| LLMLingua | 31.8 | 37.5 | 26.2 | 67.2 | 8.3 | 53.2 | 37.4 | 3,421 | 3x | 9.2(1.7x) | 30.7 | 3,366 | 3x | 7.4(1.7x) |
| **LongLLMLingua** | **40.7** | **46.2** | **27.2** | **70.6** | **53.0** | **55.2** | **48.8** | 3,283 | 3x | 8.0(1.6x) | **32.8** | 3,412 | 3x | 8.2(1.5x) |
| *2,000 tokens constraint* | | | | | | | | | | | | | | |
| *Retrieval-based Methods* | | | | | | | | | | | | | | |
| BM25 | 30.1 | 29.4 | 21.2 | 19.5 | 12.4 | 29.1 | 23.6 | 1,985 | 5x | 4.6(3.4x) | 20.1 | 1,799 | 5x | 3.8(3.2x) |
| SBERT | 33.8 | 35.9 | 25.9 | 23.5 | 18.0 | 17.8 | 25.8 | 1,947 | 5x | 4.8(3.4x) | 20.5 | 1,773 | 6x | 4.1(3.0x) |
| OpenAI | 34.3 | 36.3 | 24.7 | 32.4 | 26.3 | 24.8 | 29.8 | 1,991 | 5x | 10.4(1.5x) | 20.6 | 1,784 | 5x | 9.9(1.2x) |
| LongLLMLingua $r_k$ | 37.8 | 41.7 | 26.9 | 66.3 | 53.0 | 52.4 | 46.3 | 1,960 | 5x | 4.7(3.3x) | 24.9 | 1,771 | 6x | 10.4(1.2x) |
| *Compression-based Methods* | | | | | | | | | | | | | | |
| Selective-Context | 16.2 | 34.8 | 24.4 | 15.7 | 8.4 | 49.2 | 24.8 | 1,925 | 5x | 47.1(0.3x) | 19.4 | 1,865 | 5x | 47.5(0.3x) |
| LLMLingua | 22.4 | 32.1 | 24.5 | 61.2 | 10.4 | **56.8** | 34.6 | 1,950 | 5x | 5.9(2.6x) | 27.2 | 1,862 | 5x | 4.8(2.5x) |
| **LongLLMLingua** | **39.0** | **42.2** | **27.4** | **69.3** | **53.8** | 56.6 | **48.0** | 1,809 | 6x | 6.1(2.6x) | **32.5** | 1,753 | 6x | 5.2(2.3x) |
| Original Prompt | 39.7 | 38.7 | 26.5 | 67.0 | 37.8 | 54.2 | 44.0 | 10,295 | - | 15.6 | 32.5 | 9,788 | - | 12.2 |
| Zero-shot | 15.6 | 31.3 | 15.6 | 40.7 | 1.6 | 36.2 | 23.5 | 214 | 48x | 1.6(9.8x) | 10.8 | 32 | 306x | 1.0(12.2x) |

information density with a higher compression rate. (5) The proposed reordering method helps in not only our approach but also other baselines as shown in Table 1, well demonstrating its effectiveness.

**Ablation study** To evaluate the contributions of different components in LongLLMLingua, we introduce following variants of it for ablation study. (1) Variants about Question-aware Coarse-grained Compression, include: ours w/o Question-awareness, which calculates question-text relevance $r_k$ using information entropy in LLMLingua, ours w/ SBERT, which employs SBERT to compute $r_k$, ours w/ $p(\mathbf{x}_k^{doc}|x_i^{que,restrict})$, which replace $p(x_i^{que,restrict}|\mathbf{x}_k^{doc})$ with $p(\mathbf{x}_k^{doc}|x_i^{que,restrict})$ in Eq. 2, and ours w/o restrict, which only calculates the conditional probability corresponding to $x^{que}$. (2) Ours w/o Question-aware Fine-grained, which disregards Eq. (3) and only applies Iterative Token-level Prompt Compression as LLMLingua. (3) Ours w/o Dynamic Compression Ratio, where all documents share the same compression ratio in fine-grained compression.

Table 3: Ablation study on NaturalQuestions with 2x constraint using GPT-3.5-Turbo.

| | 1st | 5th | 10th | 15th | 20th |
|---|---|---|---|---|---|
| **LongLLMLingua** | **77.2** | **72.9** | **70.8** | **70.5** | **70.6** |
| *Question-aware Coarse-grained* | | | | | |
| - w/o Question-awareness | 42.1 | 40.3 | 39.7 | 40.1 | 40.3 |
| - w/ SBERT | 73.2 | 68.5 | 65.7 | 66.1 | 66.7 |
| - w/ $p(\mathbf{x}_k^{doc}|x_i^{que,restrict})$ | 56.0 | 52.6 | 53.4 | 51.6 | 51.1 |
| - w/o restrict | 75.1 | 72.2 | 70.3 | 70.3 | 70.2 |
| - w/o Question-aware Fine-grained | 75.8 | 71.0 | 68.9 | 68.4 | 69.3 |
| - w/o Dynamic Compression Ratio | 74.4 | 70.7 | 68.7 | 67.9 | 68.1 |
| - w/o Subsequence Recovery | 76.7 | 71.7 | 69.4 | 69.3 | 69.7 |
| - w/ Document Reordering | 76.2 | 76.2 | 76.2 | 76.2 | 76.2 |
| - w/ GPT2-small | 74.6 | 71.7 | 70.1 | 69.8 | 68.5 |
| LLMLingua | 39.7 | 39.5 | 40.4 | 37.1 | 42.3 |
| - w/ Subsequence Recovery | 43.8 | 44.1 | 43.5 | 43.3 | 44.4 |

(4) Ours w/o and (5) LLMLingua w/ Subsequence Recovery, which either removes or adds the post-processing subsequence recovery strategy. (6) Ours w/ GPT2-small, which uses the GPT2-small model as the small language model.

Table 3 shows the results of the ablation study. In summary, removing any component proposed for LongLLMLingua will lead to a performance drop regardless of the position of the ground-truth answer. This well validates the necessity and effectiveness of the proposed question-aware mechanism during coarse-to-fine compression, the dynamic compression ratio, and the subsequence recovery strategy. It also shows that applying SBERT for coarse-grained compression will result in inferior performance, which implies the superiority of our question-aware importance metric in Eq. 2 over SBERT. In addition, replacing $p(x_i^{que,restrict}|\mathbf{x}_k^{doc})$ with $p(\mathbf{x}_k^{doc}|x_i^{que,restrict})$ can greatly affect performance due to the large noise in calculating $p(\mathbf{x}_k^{doc})$ since the perplexity of document depends on many other information besides the question. Removing the restrictive statement can increase the

hallucination of small language models, leading to a decrease in performance. Moreover, our subsequence recovery strategy can also bring performance gains for LLMLingua. However, without our question-aware mechanism, results from LLMLingua are still less satisfactory. For more detailed cases, please go to Appendix F.

**Latency evaluation** We conducte end-to-end latency testing on a V100-32G, using the prompts from Multi-document QA, LongBench, and ZeroSCROLLS in the API call, and results are shown in Table 1 and 2. The latency includes the time cost for prompt compression and the request time for LLMs, with multiple measurements taken and averaged over. Results demonstrate that LongLLM-Lingua does indeed speed up the overall inference under different compression ratios and scenarios. Moreover, with the compression ratio increasing, the acceleration effect becomes more pronounced up to 2.6x. However, the OpenAI embedding and Selective-Context results in longer latency time, due to repeated API calls and the sequential entropy calculation of semantic units, respectively.

## 6 RELATED WORKS

**Long context for LLMs**. Recent research has focused on expanding the window size of LLMs. Main approaches include: (1) Staged pre-training (Nijkamp et al., 2023) which gradually increases the context window; (2) Modifying (Press et al., 2022) or interpolating position embeddings (Chen et al., 2023; Peng et al., 2023; Han et al., 2023); (3) Using linear or sparse attention mechanisms (Ding et al., 2023; Sun et al., 2023); (4) Utilizing external memory modules for context storage (Bertsch et al., 2023; Tworkowski et al., 2023). While these methods address context window expansion, their impact on downstream task performance has yet to be discussed.

**Information distribution in prompt**. Recent empirical experiments have shown that LLM performance decreases with less effective information in a prompt (Bai et al., 2023; Li et al., 2023a; Shi et al., 2023). Moreover, the position of relevant information in a prompt has a significant impact on performance(Wu et al., 2022). Liu et al. (2023) suggests that LLMs have more difficulty comprehending information located in the middle of a prompt compared to those at the edges.

**Retrieval methods** can be categorized as dense or sparse retrieval methods. Sparse retrieval methods, like BM25, determine the relevance between queries and documents based on n-gram information. Conversely, dense retrieval methods assess the relevance between queries and documents in latent space using dense vectors, such as SentenceBERT (Reimers & Gurevych, 2019) and OpenAI Embedding. Recently, Jiang et al. (2023b)) proposed an unsupervised dense retrieval method that leverages traditional compression algorithms, such as gzip, and k-nearest neighbors.

**Prompt compression methods** can be grouped into three main categories: (1) Token pruning (Goyal et al., 2020; Kim & Cho, 2021; Modarressi et al., 2022) and token merging (Bolya et al., 2023), which need model fine-tuning or intermediate results during inference and have been used with BERT-scale models. (2) Soft prompt tuning methods like GIST (Mu et al., 2023), AutoCompressor (Chevalier et al., 2023), and ICAE (Ge et al., 2023), which require LLMs' parameter fine-tuning, making them suitable for specific domains but not directly applicable to black-box LLMs. (3) Information-entropy-based approaches such as Selective Context (Li, 2023) and LLMLingua (Jiang et al., 2023a), which use a small language model to calculate the self-information or perplexity of each token in the original prompt and then remove tokens with lower perplexities.

## 7 CONCLUSION

We propose LongLLMLingua to address the three challenges, *i.e.*, higher computational/financial cost, longer system latency, and inferior performance for LLMs in long context scenarios. We develop LongLLMLingua from the perspective of efficient prompt compression, thus reducing both computational/financial cost and the system latency. We further design four components, *i.e.*, a question-aware coarse-to-fine compression method, a document reordering mechanism, dynamic compression ratios, and a post-compression subsequence recovery strategy to improve LLMs' perception of the key information, with which LongLLMLingua demonstrate superior performance. Experiments on one multi-document QA benchmark and two long context benchmarks demonstrate that LongLLMLingua compressed prompt can derive higher performance than original prompts while both API costs for inference and the end-to-end system latency are largely reduced.

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

## A  Token-level Subsequence Recovery Details

---

**Algorithm 1** Pseudo code of Token-level Subsequence Recovery.

---

**Input**: The original prompt $\boldsymbol{x}$; the compressed prompt $\widetilde{\boldsymbol{x}}$; the generation response of LLMs $\boldsymbol{y}$.
1:  Set the final response list $\boldsymbol{y}_{\text{rec}} = \phi$, the left token index of subsequence $l$ to 0.
2:  **while** $l < \boldsymbol{y}.len()$ **do**
3:      **if** Substring $y_l \in \widetilde{\boldsymbol{x}}$ **then**
4:          Find the longer substring $\widetilde{\boldsymbol{y}}_{\text{key},l} = \{y_l, y_{l+1}, ..., y_r\} \in \widetilde{\boldsymbol{x}}$.
5:          Find the maximum common shortest subsequence $\boldsymbol{x}_{i,j} = \{x_i, x_{i+1}, ..., x_j\}$ in the original prompt
    $\boldsymbol{x}$.
6:          Add the subsequence $\boldsymbol{x}_{i,j} = \{x_i, x_{i+1}, ..., x_j\}$ to the response $\boldsymbol{y}_{\text{rec}}$.
7:          Set the left index $l$ to $r + 1$.
8:      **else**
9:          Add the token $y_l$ to the response $\boldsymbol{y}_{\text{rec}}$.
10:         Set the left index $l$ to $l + 1$.
11:     **end if**
12: **end while**
**Output**: The final response list $\boldsymbol{y}_{\text{rec}}$.

---

## B  Derivation Of Question-Aware Fine-Grained Compression

Based on the definition of Eq. 3, we can derive that,

$$
\begin{aligned}
s_i &= \text{perplexity}(x_i|x_{<i}) - \text{perplexity}(x_i|x^{\text{que}}, x_{<i}) \\
&= q(x_i) \log p(x_i|x^{\text{que}}, x_{<i}) - q(x_i) \log p(x_i|x_{<i}) \\
&= q(x_i) \log \frac{p(x_i|x^{\text{que}}, x_{<i})}{p(x_i|x_{<i})}
\end{aligned}
\tag{6}
$$

In the actual calculation of perplexity, a log operation is performed to avoid overflow, and $q(x_i)$ represents the probability distribution of the ground-truth.

At the same time, we can derive the following expanded expression based on Bayes' theorem.

$$
p(x^{\text{que}}|x_i, x_{<i}) = \frac{p(x_i|x^{\text{que}}, x_{<i})p(x^{\text{que}})}{p(x_i|x_{<i})} = p(x^{\text{que}})\frac{p(x_i|x^{\text{que}}, x_{<i})}{p(x_i|x_{<i})}
\tag{7}
$$

The probability distribution $p(x^{\text{que}})$ of the question and the ground-truth distribution $q(x_i)$ of $x_i$ are constants, hence $s_i$ can be considered as the representation of Eq. 7.

$$
s_i \propto p(x^{\text{que}}|x_i, x_{<i})
\tag{8}
$$

So we can utilize Eq. 3 to represent the probability distribution $p(x^{\text{que}}|x_i, x_{<i})$, which represents the condition likelihood of generating $x^{\text{que}}$ given the token $x_i$. Therefore, we can represent the token-level sensitive distribution for the question $x^{\text{que}}$ using just a single inference. For tokens that are unrelated to $x^{\text{que}}$, such as the tokens on the right side of Figure 3b, their original amount of information may be high, but the contrastive perplexity remains at a relatively low level.

## C  Experiment Details

### C.1  Dataset Details

**NaturalQuestions multi-document QA**  A multi-document question-answering dataset, comprising 2,655 problems, was built by Liu et al. (2023) based on the NaturalQuestions dataset (Kwiatkowski et al., 2019). This dataset provides a realistic retrieval-augmented generation setup that closely resembles commercial search and question-answering applications (e.g., Bing Chat). Each example in the dataset contains a question and k related documents, utilizing the Contriever retrieval system (Izacard et al., 2022), one of which includes a document with the correct answer. To perform this task, the model must access the document containing the answer

within its input context and use it to answer the question. The dataset's data is sourced from the NaturalQuestions dataset, which contains historical queries issued to the Google search engine and human-annotated answers extracted from Wikipedia. The average prompt token length in this benchmark is 2,946. For our experiments, we used the version provided by Liu et al. (2023) that includes 20 documents[7]. The dataset comprises five different ground truth document position settings in the prompt: 1st, 5th, 10th, 15th, and 20th.

**LongBench** A multi-task long context benchmark consists of 3,750 problems in English and includes six categories with a total of 16 tasks. These tasks encompass key long-text application scenarios, such as single-document QA, multi-document QA, summarization, few-shot learning, synthetic tasks, and code completion. The average prompt token length in this benchmark is 10,289. For our experiments, we used the English dataset and evaluation scripts provided by Bai et al. (2023) for this benchmark[8].

**ZeroSCROLLS** The multi-task long context benchmark consists of 4,378 problems, including four categories with a total of 10 tasks. These tasks cover summarization, question answering, aggregated sentiment classification, and information reordering. The average prompt token length in this benchmark is 9,788. For our experiments, we used the validation set and evaluation scripts provided by Shaham et al. (2023) for this dataset[9].

**MuSiQue** The multi-hop question-answer dataset is composed of 39,876, 4,834, and 4,918 problems in the training, validation, and testing datasets, respectively. This dataset requires the language model to conduct multiple inferences based on the content of several documents and provide corresponding answers, thereby necessitating a certain capability for global information processing. The average token length for prompts in this dataset is 2,477. For our experiments, we utilized the validation set and evaluation scripts provided by Trivedi et al. (2022) for this dataset[10].

**LooGLE** The multi-task long context benchmark comprises 6,448 problems, divided into three categories: summarization, short dependency question answering, and long dependency question answering. The average prompt token length in this benchmark stands at 24,005. For our experiments, we focused on the long dependency question answering subset, which includes four types of tasks: information retrieval, timeline reordering, computation, and comprehension. This subset contains 1,101 problems. We utilized the evaluation scripts provided by Li et al. (2023b) for this dataset[11].

### C.2 OTHER IMPLEMENTATION DETAILS

All experiments were conducted using a Tesla V100 (32GB). We use tiktoken[12] and GPT-3.5-Turbo model to count all the tokens. We set the granular control coefficient $k$ to 2. We use the pre-defined compression rates $\tau_{\text{ins}} = 0.85$ and $\tau_{\text{que}} = 0.9$ for instructions and questions. The segment size used in the iterative token-level compression is set to 200. The $\delta\tau$ used in dynamic compression ratio is set to 0.25. For a fair comparison, we only used reordering in the NaturalQuestions Multi-document QA and noted this in Table 1. We use "*We can get the answer to this question in the given documents.*" as the guideline sentence in Equation (3).

For the baselines experiment, we use the currently recommended strongest model, all-mpnet-base-v2[13], as the dense representation model for SentenceBERT. We use the recommended "text-embedding-ada-002" as the embedding model for OpenAI Embedding[14]. We use the GPT2-dolly[15] as the small language model in w/ GPT2-small ablation experiments.

---

[7]https://github.com/nelson-liu/lost-in-the-middle

[8]https://github.com/THUDM/LongBench

[9]https://www.zero.scrolls-benchmark.com/

[10]https://github.com/stonybrooknlp/musique

[11]https://github.com/bigai-nlco/LooGLE

[12]https://github.com/openai/tiktoken

[13]https://www.sbert.net/docs/pretrained_models.html

[14]https://platform.openai.com/docs/guides/embeddings/

[15]https://huggingface.co/lgaalves/gpt2-dolly

# D   ADDITIONAL EXPERIMENTAL RESULTS

## D.1   DOCUMENT-LEVEL AVERAGE PERPLEXITY DISTRIBUTION

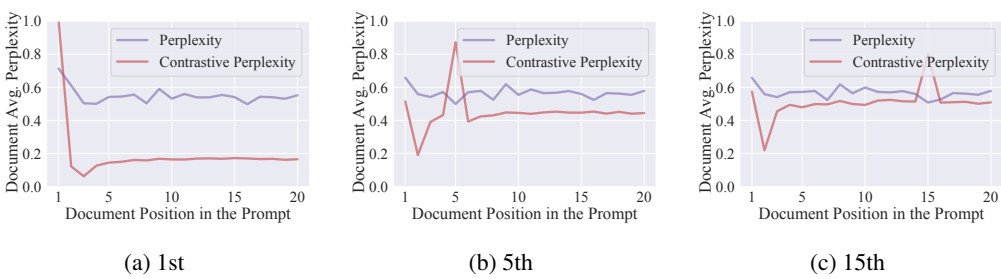

(a) 1st                     (b) 5th                     (c) 15th

Figure 5: The distribution of document-level average perplexity when the ground-truth document is in different positions.

Figure 5 shows the distribution of the document's average perplexity when the ground-truth is located at different positions within the prompt. As can be observed, as the context length increases, the original perplexity curve remains relatively stable. In unrelated documents, a higher perplexity is still retained, making it easier to remove relevant tokens from the related documents in the prompt compression process, thereby damaging the corresponding semantic information. Contrarily, contrastive perplexity shows an increase in perplexity in documents related to the question. According to the theoretical derivation in Appendix B, it's known that contrastive perplexity characterizes the conditional probability of tokens corresponding to the question. The higher the relevance, the higher the contrastive perplexity, thereby retaining key information in the prompt compression process.

## D.2   ZEROSCROLLS BREAKDOWNS

Table 4: Performance breakdown of different methods under different compression ratios on Zero-SCROLLS (Shaham et al., 2023) using GPT-3.5-Turbo.

| Methods | GvRp | SSFD | QMsm | SQAL | QALT | Nrtv | Qspr | MuSQ | SpDg | BkSS | AVG | Tokens | $1/\tau$ |
|---|---|---|---|---|---|---|---|---|---|---|---|---|---|
| *3,000 tokens constraint* | | | | | | | | | | | | | |
| *Retrieval-based Methods* | | | | | | | | | | | | | |
| BM25 | 9.7 | 3.4 | 11.7 | 14.3 | 57.1 | 5.9 | 25.7 | 11.2 | 29.6 | 29.6 | 19.8 | 3,379 | 3x |
| SBERT | 16.5 | 9.8 | 12.3 | 15.2 | 60.0 | 14.6 | 23.4 | 12.1 | 39.4 | 36.4 | 24.0 | 3,340 | 3x |
| OpenAI | 14.3 | 8.3 | 12.0 | 15.3 | 66.7 | 13.3 | 24.3 | 11.7 | 31.2 | 26.4 | 22.4 | 3,362 | 3x |
| LongLLMLingua $r_k$ | 19.5 | 11.6 | 14.7 | 15.5 | 66.7 | 20.5 | 27.6 | 13.0 | 60.8 | 43.4 | 29.3 | 3,350 | 3x |
| *Compression-based Methods* | | | | | | | | | | | | | |
| Selective-Context | 20.8 | 9.1 | 11.7 | 13.4 | 50.0 | 9.8 | 26.1 | 11.0 | 46.0 | 9.5 | 20.7 | 3,460 | 3x |
| LLMLingua | 18.7 | 10.0 | 14.9 | 16.8 | 61.9 | 26.9 | 27.2 | 23.4 | 62.9 | 44.5 | 30.7 | 3,366 | 3x |
| **LongLLMLingua** | 21.9 | 12.7 | 15.5 | 17.0 | 66.9 | 27.6 | 31.1 | 23.8 | 65.6 | 46.4 | 32.8 | 3,412 | 3x |
| *2,000 tokens constraint* | | | | | | | | | | | | | |
| *Retrieval-based Methods* | | | | | | | | | | | | | |
| BM25 | 8.8 | 2.5 | 11.1 | 13.5 | 60.0 | 7.0 | 4.9 | 20.3 | 39.9 | 32.9 | 20.1 | 1,799 | 5x |
| SBERT | 10.2 | 7.9 | 13.7 | 13.2 | 60.0 | 8.1 | 10.8 | 1.7 | 37.2 | 42.8 | 20.5 | 1,773 | 6x |
| OpenAI | 11.1 | 8.0 | 11.8 | 13.6 | 60.0 | 7.1 | 13.2 | 4.0 | 33.6 | 43.6 | 20.6 | 1,784 | 5x |
| LongLLMLingua $r_k$ | 18.2 | 9.8 | 12.3 | 15.9 | 57.1 | 10.1 | 17.8 | 7.3 | 57.7 | 42.3 | 24.9 | 1,771 | 6x |
| *Compression-based Methods* | | | | | | | | | | | | | |
| Selective-Context | 19.0 | 8.4 | 9.7 | 12.4 | 47.0 | 12.5 | 21.6 | 11.5 | 41.2 | 11.0 | 19.4 | 1,865 | 5x |
| LLMLingua | 19.4 | 11.9 | 13.1 | 16.0 | 62.1 | 23.7 | 24.0 | 22.4 | 33.9 | 44.9 | 27.2 | 1,862 | 5x |
| **LongLLMLingua** | 19.9 | 12.3 | 14.7 | 16.5 | 64.9 | 27.4 | 30.6 | 23.5 | 68.3 | 47.1 | 32.5 | 1,809 | 6x |
| Original Prompt | 21.8 | 12.1 | 17.9 | 17.4 | 66.7 | 25.3 | 29.8 | 20.0 | 69.7 | 44.1 | 32.5 | 9,788 | - |
| Zero-shot | 9.4 | 3.0 | 8.6 | 11.4 | 42.9 | 10.6 | 12.4 | 5.5 | 4.2 | 0.0 | 12.8 | 32 | 306x |

Table 4 presents a detailed performance breakdown on the ZeroSCROLLS benchmark. It can be observed that in the four summarization tasks - GvRp, SSFD, QMsm, SQAL, LongLLMLingua closely matches or slightly surpasses the original results under two compression constraints. Meanwhile, in the four long context QA tasks - Qsqr, Nrtv, QALT, MuSQ, there is a significant improvement. Notably, in the MuSiQue task, which is based on a question-answering dataset from books and movie scripts, there is a 2.1 point increase even under a 2,000 tokens constraint. It's worth mentioning that MuSiQue is a multi-hop question-answering dataset that requires LLMs to utilize global information for long dependency QA. LongLLMLingua can also improve by 3.5 points under a 6x compression ratio. In the two ordering tasks, SpDg and BkSS, LongLLMLingua can better retain globally sensitive information, resulting in a 3.0 point improvement in BkSS after prompt compression.

It's important to note that although the ZeroScrolls validation dataset is relatively small, it still demonstrates conclusions similar to previous experimental observations across various methods and tasks. Furthermore, this study conducted an in-depth analysis of the multi-hop QA task - MuSiQue, and another long context benchmark - LooGLE. The results can be found in Appendix D.3 and Appendix D.6.

## D.3 MUSIQUE

Table 5 presents the results from the MuSiQue multi-hop question-answer dataset. From the table, it can be observed that in the multi-hop QA task, requiring global information: 1) LongLLMLingua can reduce noise in the prompt by eliminating irrelevant information and putting more related information at the beginning or end of the prompt, thereby improving performance by 5.4 points. 2) The performance drop is more pronounced for retrieval-based methods, particularly for n-gram-based methods like BM25. Due to long dependencies, direct matching information is lost, resulting in less relevant information being recalled. 3) The performance of compression-based methods is slightly different. Selective-Context does not distinguish between different modules' sensitivity, resulting in a loss of question and instruction-related information, thereby leading to poorer performance. However, LLMLingua can still retain relevant key information at around a 2x compression ratio. 4) The ablation experiments show that every module designed in LongLLMLingua plays a role in the multi-hop task. The removal of the question-aware coarse-grained and w/ $p(\mathbf{x}_k^{\mathrm{doc}}|x_i^{\mathrm{que,restrict}})$ modules, which have dif-

Table 5: Performance of different methods and ablation study on MuSicQue (Trivedi et al., 2022) with 2x constraint using GPT-3.5-Turbo.

| Methods | F1 | Tokens | $1/\tau$ |
|---|---|---|---|
| Original Prompt | 45.8 | 2,427 | - |
| BM25 | 28.5 | 1,295 | 1.9x |
| SBERT | 36.2 | 1,288 | 1.9x |
| LongLLMLingua $r_k$ | 46.3 | 1,295 | 1.9x |
| Selective-Context | 19.6 | 1,141 | 2.1x |
| LLMLingua | 40.1 | 1,110 | 2.2x |
| **LongLLMLingua** | **51.2** | 1,077 | 2.3x |
| *Question-aware Coarse-grained* | | | |
| - w/o Question-awareness | 43.2 | 1,076 | 2.3x |
| - w/ SBERT | 47.3 | 1,070 | 2.3x |
| - w/ $p(\mathbf{x}_k^{\mathrm{doc}}|x_i^{\mathrm{que,restrict}})$ | 44.0 | 1,066 | 2.3x |
| - w/o restrict | 49.2 | 1,078 | 2.3x |
| - w/o Question-aware Fine-grained | 48.4 | 1,118 | 2.2x |
| - w/o Dynamic Compression Ratio | 48.2 | 1,090 | 2.2x |
| - w/o Subsequence Recovery | 50.7 | 1,077 | 2.3x |
| - w/o Document Reordering | 49.2 | 1,077 | 2.3x |
| - w/ GPT2-small | 48.4 | 1,095 | 2.2x |

ficulty in perceiving the importance distribution of corresponding questions, can cause a drop of up to 8 points. Removing the restrict prompt in the question-aware coarse module can also cause a 2-point drop due to the hallucination issue of small LLM. In addition, removing question-aware fine-grained, dynamic compression ratio, and document reordering can all cause a drop of 0.5-2.8 points. 5) Moreover, if the small language model in LongLLMLingua is replaced with GPT2-small, it can further improve the acceleration ratio and still achieve a result that is 2.6 points better than the original prompt.

## D.4 ABLATION IN LONGBENCH

Table 6 presents the results from the ablation experiment in the LongBench long context benchmark. It can be observed that in various long context tasks: 1) Removing the question-aware coarse-grained, question-aware fine-grained, dynamic compression ratio, document reordering, and subsequence recovery proposed by LongLLMLingua all result in different degrees of performance drop. 2) Among these, question-aware coarse-grained is particularly important for document-based QA and synthetic tasks, with the maximum drop being 35.8 points; its impact on summarization and

Table 6: Ablation on LongBench (Bai et al., 2023) using GPT-3.5-Turbo.

| Methods | SingleDoc | MultiDoc | Summ. | FewShot | Synth. | Code | AVG | Tokens | $1/\tau$ |
|---|---|---|---|---|---|---|---|---|---|
| **LongLLMLingua** | 39.0 | 42.2 | 27.4 | 69.3 | 53.8 | 56.6 | 48.0 | 1,809 | 6x |
| *Question-aware Coarse-grained* | | | | | | | | | |
| - w/o Question-awareness | 27.1 | 38.7 | 25.4 | 62.0 | 18.0 | 53.3 | 37.4 | 1,945 | 5x |
| - w/ SBERT | 34.0 | 38.7 | 24.1 | 57.9 | 32.5 | 31.1 | 36.4 | 1790 | 6x |
| - w/ $p(\mathbf{x}_k^{\text{doc}}|x_i^{\text{que,restrict}})$ | 22.5 | 28.9 | 23.2 | 53.0 | 22.5 | 33.3 | 30.6 | 1,794 | 6x |
| - w/o restrict | 37.8 | 39.5 | 26.4 | 64.8 | 52.5 | 55.8 | 46.1 | 1,834 | 6x |
| - w/o Question-aware Fine-grained | 35.7 | 41.1 | 26.4 | 62.9 | 44.5 | 54.8 | 44.2 | 1,807 | 6x |
| - w/o Dynamic Compression Ratio | 36.1 | 40.6 | 26.9 | 67.2 | 48.0 | 55.8 | 45.7 | 1,851 | 6x |
| - w/o Subsequence Recovery | 38.6 | 41.8 | 27.3 | 69.0 | 53.8 | 56.6 | 47.8 | 1,809 | 6x |
| - w/ Document Reordering | 39.9 | 43.2 | 27.4 | 69.8 | 53.0 | 56.7 | 48.3 | 1,822 | 6x |
| - w/ GPT2-small | 35.9 | 39.4 | 25.0 | 60.6 | 42.0 | 55.4 | 43.0 | 1,892 | 5x |

code tasks is relatively smaller. 3) The design of the conditional probability in the question-aware coarse-grained module improves the results in all tasks, including code completion, single-document question-answer, and synthetic tasks. Changing the order of conditional probabilities or removing the restrict prompt both lead to varying degrees of performance decline. 4) Removing question-aware fine-grained, dynamic compression ratio has a more significant impact on document-based QA and synthetic tasks. 5) The subsequence recovery module can enhance reference-based tasks, but its improvement on tasks like summarization, code, synthetic, etc., is relatively smaller. 6) Document reordering is effective for all types of tasks. Reordering at the document level does not affect LLMs' understanding of context information, even for timeline-related tasks (see timeline reorder in LooGLE, Table 8). On the contrary, reordering can effectively alleviate the "lost in the middle" issue, thereby improving LLMs performance. 7) Using GPT2-small reduces the capture of effective tokens, but it can still achieve results close to or even slightly better than the original prompt.

## D.5   LONGBENCH USING LONGCHAT-13B-16K

Table 7: Performance of different methods under different compression ratios on LongBench (Bai et al., 2023) using LongChat-13b. Considering the dataset structure, we do not use the reordering strategy here.

| Methods | SingleDoc | MultiDoc | Summ. | FewShot | Synth. | Code | AVG | Tokens | $1/\tau$ |
|---|---|---|---|---|---|---|---|---|---|
| Original Prompt | 27.4 | 30.3 | 20.3 | 49.9 | 12.5 | 42.5 | 30.5 | 10,295 | - |
| *Retrieval-based Methods* | | | | | | | | | |
| BM25 | 2.4 | 2.6 | 16.4 | 8.7 | 0.0 | 44.7 | 12.5 | 1,985 | 5x |
| SBERT | 11.6 | 13.7 | 21.1 | 16.2 | 7.5 | 30.0 | 16.7 | 1,947 | 5x |
| LongLLMLingua $r_k$ | 30.3 | 32.4 | 24.5 | 41.0 | 27.5 | 38.1 | 32.3 | 1,960 | 5x |
| *Compression-based Methods* | | | | | | | | | |
| Selective-Context | 16.1 | 23.5 | 21.8 | 21.4 | 2.5 | 35.9 | 20.2 | 1,925 | 5x |
| LLMLingua | 20.6 | 22.3 | 22.4 | 35.6 | 0.0 | 35.4 | 22.7 | 1,950 | 5x |
| **LongLLMLingua** | 31.1 | 34.1 | 24.5 | 45.7 | 28.0 | 48.6 | 35.3 | 1,809 | 6x |

Table 7 presents the experiment results in the LongBench long context benchmark using LongChat-13b-16k. It can be seen that the compressed prompt can also achieve good results on other LLMs, such as LongChat-13b-16k. Specifically, 1) there is a maximum improvement of 15.5 points in synthetic tasks. Except for a slight drop in few-shot Learning, there is an improvement of 3-5 points in other tasks. 2) The performance trends of retrieval-based and compressed-based baselines are similar to the results in GPT-3.5-Turbo.

## D.6   LOOGLE

Table 8 presents the experiment results in the LooGLE long dependency benchmark, which features longer prompts (~30k) and more global dependencies. From the table, we can observe that: 1) LongLLMLingua can effectively improve the performance of long context tasks by compressing

Table 8: Performance of different methods on LooGLE (Li et al., 2023b) long dependency QA.

| Methods | Retrieval | Timeline Reorder | Computation | Reasoning | AVG | Tokens | $1/\tau$ |
|---|---|---|---|---|---|---|---|
| *Retrieval-based Methods* | | | | | | | |
| BM25 | 20.4 | 21.7 | 8.2 | 26.3 | 19.2 | 3,185 | 10x |
| SBERT | 28.9 | 21.1 | 10.7 | 27.2 | 22.0 | 3,169 | 10x |
| LongLLMLingua $r_k$ | 38.6 | 32.2 | 16.2 | 26.3 | 28.3 | 3,158 | 10x |
| *Compression-based Methods* | | | | | | | |
| Selective-Context | 16.7 | 5.0 | 2.3 | 17.6 | 10.4 | 3,710 | 8x |
| LLMLingua | 10.0 | 25.0 | 13.3 | 21.1 | 17.3 | 3,404 | 9x |
| **LongLLMLingua** | **40.0** | **35.0** | **19.7** | **33.6** | **32.1** | 3,121 | 10x |
| **LongLLMLingua** w/o Reorder | 39.3 | 33.8 | 18.7 | 31.6 | 30.9 | 3,119 | 10x |
| Original Prompt | 24.1 | 20.9 | 13.5 | 32.1 | 22.6 | 30,546 | - |
| Zero-shot | 8.7 | 6.3 | 1.2 | 14.5 | 7.7 | 43 | 710x |

prompts, even for long dependency tasks. The results show that LongLLMLingua significantly improves performance in tasks such as retrieval, timeline reorder, and computation, with the maximum improvement reaching 15.9 points. 2) The document reorder in LongLLMLingua is effective in all types of tasks, even in tasks highly related to the timeline, it can effectively improve performance by alleviating the "lost in the middle" issue. 3) Retrieval-based methods tend to lose performance in tasks that have longer dependencies, such as computation and reasoning. 4) For compression-based methods, due to the difficulty in perceiving question information, there tends to be a larger performance loss in retrieval tasks within long contexts.

# E ECONOMIC COST

Table 9: The inference costs(per 1,000 samples $) for various datasets using GPT-3.5-Turbo.

| | Multi-document QA | LongBench | ZeroScolls | MuSicQue | LooGLE |
|---|---|---|---|---|---|
| Original | 4.6 | 31.5 | 30.6 | 3.8 | 93.6 |
| Ours | 1.3 | 3.0 | 3.2 | 1.8 | 5.6 |

Table 9 presents the estimated per 1,000 samples inference costs for various datasets, encompassing input prompts and generated output text, based on GPT-3.5-Turbo pricing[16]. Our approach demonstrates substantial savings in computational resources and monetary expenses, particularly in long context situations. Cost reductions of $3.3, $28.5, $27.4, $2.0, and $88.0 per 1,000 samples are observed for Multi-document QA, LongBench, ZeroScrolls, MuSiQue, and LooGLE, respectively.

# F ABLATION ANALYSIS

# G CASES STUDY

---

[16]https://openai.com/pricing

**Ours w/o Token-level Question-aware:**
**Compressed Prompt:**
Write a high-quality answer for the given question using only the provided search results (some of which might be irrelevant).
Document [1](: Physics)gen,, who received2K, which is ,73,0 in0. Johnen only to twice6. Mariaie won, for.g was, until1estate he. Two:Mayer (1963). As of 2017, the prize has been awarded
Question: who got the first nobel prize in physics
Answer:
**LLMs' Response:**
No answer found in the given search results.

- - - - - - - - - - - - - - - - - - - - - - - - - - - - - - - - - - - - - - - - - - - - - - -

**Ours w/ Token-level Question-aware:**
**Compressed Prompt:**
Write a high-quality answer for the given question using only the provided search results (some of which might be irrelevant).
1Title: List of Nobelates in The first Nobel Prize was1 to $\boxed{Wilhelmrad}$, of who received 1582 which,70 in0 en the prize. Skska also won two Nobeles for physics3g01, theate he women prize:ertMayer (1963). As of 2017, the prize has been awarded
Question: who got the first nobel prize in physics
Answer:
**LLMs' Response:**
Wilhelmrad
**LLMs' Response after Subsquence Recovery:**
Wilhelm Conrad Röntgen
**Ground Truth:**

Wilhelm Conrad Röntgen

Figure 6: Comparing the compressed prompt and LLMs' response before and after using Question-aware Fine-grained Compression and Subsequence Recovery($1/\tau$=30x, high compression ratio setting) from NaturalQuestions Multi-document QA (Liu et al., 2023) using GPT-3.5-Turbo.

**Original Prompt:**

...

Document [1](Title: Dancing on Ice) It was confirmed on 25 January 2018, that Dancing on Ice had been recommissioned for an eleventh series to air in 2019.

...

**Compressed Prompt:**

Write a high-quality answer for the given question using only the provided search results (some of which might be irrelevant).

1Title: Dancing on was confirmed on 2 January 2018 that Dancing on had been recommissioned for an eleventh series air in 209 .

Document [2Title: Dan on) Dan on Ice Dancing on British presented by Phillip Schof alongside Holly Willough from 26 to 2011, and Christine Bleakley from 2012 to 204 The show consists of celebrit and professional partners figure skating in front of a panel of judges The, broadcast on ITV, started on January 2006 and ended on 9 March 2014 after showćontract not renewed by ITV On 4 September 2017, it was announced that rev series would on I 7 January 201 Sch and Willby returning as a

5(: on ( on () The third series of a from January to168TV. The from Saturdays, with Holby present Kar,y Sliner Robin Cins returned to Panel", with Ruth H joining the panel as replacement for Natalia Bestova. The commission of the was confirmed by at the07 announcedova depart the series Robinen Bar,ater and Jasoniner announced

7( on ( )) Dan 2 second of Dan on a from January to1207 ITV It presented Phillip Sch Holly Willough, and judged the "I P consisting Nicky Slater, Nataliaian Karenres Jason Gardiner Karen Barber and Robin Cousins Jaynevill and Christopher Dean co and trained the contestants In this series, cele to ten in first series. The series was won former Kyran Bracken, with Mel Lambert the winner. It announced thatenresge

Document [](  on Ice on 08 on TV edition started 8 TV2 The Russian version "анду) being on channel0, and renamed in8 to " Ice" (). Its counterpart called "Ice Age (, "Stars on Ice on Channel Oneak IceHviezdyĬJ. The Turkish version" is called Dans" ("ance on

Document1 on Ice its, all,é () and Sje Chris de In series.2 edition

](: on Ice world) Dan Ice is a made competition world format, and been subsequently Italy Chile where titled after series There have a, the show was broadcast on Channel 13 as a

Document [17](Title: Dancing on Ice) the insight to the training of the celebrities over the last week. It was presented by television presenter Ben Shephard and former contestant and "Loose Women" star Coleen Nolan. The show was broadcast from 8 pm to 8.30 pm on Friday evenings on ITV throughout the duration of the main shows season. STV who broadcast the main show did not broadcast this on the Friday evening but after repeating the previous weekś main show on the following Saturday afternoon. Due to poor ratings, "Dancing on Ice Friday" was axed prior to the 2011 series. The show was based in the

Question: when is dancing on ice on the tv

Answer:

**LLMs' Response:**

209

**LLMs' Response after Subsquence Recovery:**

2019

**Ground Truth:**

2019

Figure 7: Cases study on NaturalQuestions Multi-document QA dataset (Liu et al., 2023) in 4x constraint using GPT-3.5-Turbo.

**Compressed Prompt:**
Please complete the code given below.

```
public class MessageArchiveManagement
    private static final long MILLISECONDS_IN_DAY = 24 * 00 *0;
    public static final long_CUP = MCON_DAY
    /.../
           .("",.getStart
           add
 ifget() >0
           Node end("
            end.("
            endNode.Value("", Util.getTimestamp(query.getEnd
addNode
        }          if (.withid null && contact null && !isference
           Node with("            .with
           .Value("valuewith
           .(
        //    queryMessageive(connection, nextQuery
           final(connectionProtocol(), query
           synchronized (eries)
           //    queries.add(nextQuery
           }
        }

    public boolean queryInProgress( contact, OnLoaded
    moreMessagesLoadedListener)
       ized (eries)
           (Query query : queries)
               if(query.getWith().equals(contact.getUserId()))
    if (query.onMoreMessagesLoaded == null &&MessagesListener
    null) query.setOnMoreMessagesLoaded(Listener}
                   return true;}}
           return false;}}
    private void finalizeQuery(Protocol protocol, Query query)
       synchronized (queries) {
           .remove(query);
       }
       Contact contact = null;
       if (query.getWith() != null) {
           contact = protocol.getItemByUID(query.getWith());
       }
       if (contact != null) {
```

Next line of code:
**LLMs' Response:**

```
       contact.setLastMessageTransitted(query.getEnd());\n
```

**Ground Truth:**

```
       if (contact.setLastMessageTransmitted(query.getEnd()))
```

**Zero-shot LLMs' Response:**

```
       contact.removeQuery(query);\n
```

Figure 8: Cases study on lcc code completion task in LongBench benchmark (Bai et al., 2023) in 2,000 constraint using GPT-3.5-Turbo.

**Compressed Prompt:**

Please the of the question. questions

are sometimes your cold but the of you isnt:ason: What food hasges:: Who the first coach the Clevelandns What arch the Placede: Other: Who created Harryime What Carbean cult didvey:: did Iraqi troops::ose cover is of an of Universal Import What the of Betty theest thectic:: Wh the founder and of The National Review:: was T Tims What the historicalals following the of Agra is whiteolate: of What the the: is a of everything:ase and:ose old London come- was : "y my sweet:: The major team in is called: Group or organization of: How dorow: M of: the name to ofese ?: Animal: is gymnia: of the between k and ch: of: the lawyer for Randy C:: the Francisco What year the in whereci became What country most is g the Who the to P What are the states the the name , Elino: What manmade waterways is1.76: Other of Z:ivalent of: of What was the:: How do ants have: of: the Dow first the high sound that hear in ear every then , but then it away ,:: didist control in:: How can I ofies ' What did theramid-ers of Egypt eat:: How does Belle her inast: M of: When reading classs does EENTY :: Expression abbre: When was Florida:: manyelies were killed the: Whative on Punchl Hill and has1 What the Filenes the cookies in Internet: What word contains: Word with a special is Larry: a person: a Frenchist: of What American wrote : " Goodors:: Where theiestk rail stations:: many people ofosis: the worsticane Whatbean is of was Jean: What the2 What caused Harryini What buildingately enough the the1d bill: Other location: many logmic there a rule:: the the word , JJ the average hours per months byOL:: How a cop of: many are of is Ch:: is Whatation does: the the Whatte is " a whole new: Other: the Chyl nuclear:

the first the: Invention, book and otherative What does " Philebus-:: didoco painting: the between: is Po What. the lowest highestation 6:: How the inpy: an the " What was General Douglasthur in was by Presidentuman: How isaster: an the forini:: was Dick:: Where can find on religion and health the and: Other Whatian the TV51 theBC show for How the is of What Englishrighted " thee , so What song put James:ative piece

What new school in Philadelphia: Whatwestern isbed is B: is What Asian was as The Little Brown theans What of thean meeting: is: much the91 ?:: On which isbor: Who first:: the:: How you a paint: an What then-der theterset ,:ivalent What is to hold the lens the the star: Why toason

a for behavior , or that the accepted of:ivalent of Perg What religion What country you the What does V:: Where I a goodboard for:: buyies on the the the: areter cookiespped with cres: theoe thated ofasticitations , as ' the rules to ": the three What do for an:: CNN in:: is a:ose special bears was on17 the Who used Au an electionan: what book: is to the various ways can measure IT:chni and method is software What British minister and wereins: aic the to overcome fear What drink would the biggest:: the States do people longest:: which the the rare disease as : , andentizations , , and is of a is What Russian mastery What a perfect a: What c was Thomas in: Other: did the of What did What can feature the different:ques the-O the ons lips at What anetic did Victoria used her child: D What do: many from to of ofors , body: and is What causes get in: the G What is Other Who the1 century-stone who gained of Florence but endedake:

of c: the oldest relationship sister with The the world of a to detectchni Whaty make:: Stuart is first: is w What a character by Rs

...

Question: What is a fuel cell ?

Type:

**LLMs' Response:**

Definition of something

**LLMs' Response after Subsquence Recovery:**

Definition of something

**Ground Truth:**

Definition of something

Figure 9: Cases study on trec few-show learning in LongBench benchmark (Bai et al., 2023) in 2,000 constraint using GPT-3.5-Turbo.

