# OpenReview forum: "LongLLMLingua: Accelerating and Enhancing LLMs in Long Context Scenarios via Prompt Compression"
_ICLR.cc/2024/Conference — Submitted to ICLR 2024_

### Official Review · Reviewer_jRw6 · 2023-10-28

**Soundness:** 2 fair
**Presentation:** 2 fair
**Contribution:** 2 fair
**Rating:** 6
**Confidence:** 4

**Summary:**

The paper presents LongLLMLingua, a prompt compression method to improve the high computation cost, longer latency and inferior performance in current long context LLMs.

**Strengths:**

1. the use of compression is well motivated.
2. the use of reordering mechanism is a promising method to solve the lost in the middle phenomenon.
3. the evaluation is throughout and comprehensive, especially Table 1 provides a fair and informative comparison, by using Chat-gpt and LongChat as target models (representative from close-sourced and open-sourced family).

**Weaknesses:**

1. The latency evaluation setup is not well presented. The latency of API calls may have a high variance. Are the experiments conducted several times, and in different hours? If not, the reviewer suggests improving this and update the evaluation setup.

**Questions:**

Please address the weakness above.

---

> ### Author Response · Authors · 2023-11-21
> **Response to Reviewer jRw6**
>
> 1. **"… The latency of API calls may have a high variance. Are the experiments conducted several times, and in different hours? If not, the reviewer suggests improving this and update the evaluation setup."**
> Thank you for your suggestion. We appreciate your concern about the variance in API call latency. We want to assure you that the latency reported in the paper is the result of multiple tests. However, to provide a more comprehensive latency comparison, we have added end-to-end latency results of different methods on Multi-document QA, LongBench, and ZeroSCROLLS in Table 1 and Table 2. Our results show that compared to the original prompts, LongLLMLingua can speedup the end-to-end system latency from 1.4x to 2.3x. We hope that these additional results will address your concerns.

---

> > ### Comment · Reviewer_jRw6 · 2023-11-21
> >
> > Thank you for answering this question! I have no further concerns and recommend accept this paper.

---

### Official Review · Reviewer_qdfA · 2023-10-31

**Soundness:** 3 good
**Presentation:** 3 good
**Contribution:** 3 good
**Rating:** 6
**Confidence:** 3

**Summary:**

This paper introduces LongLLMLingua as an innovative solution for prompt compression aimed at reducing costs and latency. LongLLMLingua stands out for its ability to consider the content of a question during compression, focusing on key tokens to enhance performance in scenarios with lengthy contexts. To achieve this, the authors propose several significant contributions: a question-aware compression framework, a document reordering based on a newly proposed importance metric, dynamic compression ratios, and a post-compression recovery strategy.

**Strengths:**

I appreciate the logical structure and clarity of this paper. The authors present their motivations compellingly, and the proposed LongLLMLingua method is both intuitive and seemingly effective, as evidenced by the strong results reported.

**Weaknesses:**

I have several questions and comments regarding the methods outlined in the paper that I hope the authors can address:

1. The foundational work upon which this research is built, LLMLingua, should be appropriately cited from [https://arxiv.org/abs/2310.05736](https://arxiv.org/abs/2310.05736), where the authors are clearly identified. Citing it with "Anonymous" authors seems unusual.

2. It might be beneficial to consolidate the list of contributions into three primary ones. Several contributions, such as document reordering, information reduction loss ... etc (discussed in Sections 4.2-4.4), seem relatively minor individually. They might be more appropriately categorized as a set of techniques or "tricks" that collectively enhance the framework's performance and stability.

3. Figure 5 alone seems insufficient to establish the full effectiveness of the framework. To provide a more comprehensive understanding, Tables 1 and 3 should include E2E runtime data, particularly to offer perspective on the performance of retrieval-based methods.

4. The contrastive perplexity-based importance metric's effectiveness isn't entirely convincing. Figure 3 (b) presents just a single example from a QA dataset where key documents are positioned at the beginning. Wouldn’t it be more informative to compare the average contrastive perplexity of key documents against that of nonessential documents? Similarly, a comparison using standard perplexity might also be illuminating.

5. I believe there might be a typographical error: 'ground-true' should likely be corrected to "ground-truth."

**Questions:**

see weaknesses

---

> ### Author Response · Authors · 2023-11-21
> **Response to Reviewer qdfA**
>
> 1. **"… LLMLingua... Citing it with Anonymous authors seems unusual."**
> Thank you for your suggestion, we have fixed this issue in the updated version.
>
> 2. **"It might be beneficial to consolidate the list of contributions into three primary ones."**
> Thank you for your suggestion, we will rewrite this section in subsequent versions.
>
> 3. **"Figure 5 alone seems insufficient to establish the full effectiveness of the framework... Tables 1 and 3 should include E2E runtime data, particularly to offer perspective on the performance of retrieval-based methods."**
> Thanks for the insightful feedback. we have incorporated the end-to-end latency results of different methods on Multi-document QA, LongBench, and ZeroSCROLLS in Table 1 and Table 2. The results show that while LongLLMLingua has a slightly higher latency compared to retrieval-based methods like BM25, it still accelerates the end-to-end system latency by 1.4x-2.3x when compared with the original prompts.
>
> 4. **"'Ground-true' should likely be corrected to 'ground-truth.'"**
> Thanks. We have corrected this in the updated version.

---

> > ### Comment · Reviewer_qdfA · 2023-11-22
> > **Responding to author reply**
> >
> > Thank you for responding to my questions/concerns. I will maintain my score.

---

### Official Review · Reviewer_PhaL · 2023-11-01

**Soundness:** 3 good
**Presentation:** 3 good
**Contribution:** 3 good
**Rating:** 6
**Confidence:** 5

**Summary:**

This paper introduces a technique, LongLLMLingua, to accelerating and enhancing LLMs in Long-context via compression. The method involves several aspects, coarse-to-fine compression, reordering mechanism to reduce information loss in the middle. adaptive compression control and post-compression recovery strategy. Experiments are based upon GPT-3.5 api and LongChat-13B-16k models. It presents results on several benchmarks and show the effectiveness of the proposed method.

**Strengths:**

1. The method is detailed and comprehensive. It includes many aspects that can be helpful for the compression and long-context prompting.

2. The compression ratio is great. It introduces 2x-10x compression rate and 1.4x-3.8x latency speed-up. It is very promising.

3. Results on benchmarks are good as shown in Table 1, 2, and 3.

4. The visualization in Figure 2 is very clear for me to understand this method.

**Weaknesses:**

1. In the methods, it contains several aspects coarse compression, fine compression, reordering mechanism to reduce information loss in the middle. adaptive compression control and post-compression recovery strategy. It seems that it lacks a detailed ablation study that the influence on the compression rate and performance from each aspect. This is important for us to have a better understanding of this paper.

2. For the reordering method to avoid the lost-in-the-middle issue, I have a question that whether the reordering will disturb the time-line between each document. For example, if the input documents are some sections in a fiction, the latter sections depend on the content of the former ones to understand. If so, whether this operation will disturb or introduce other difficulties for understanding?

3. Based on Figure3b, it shows that contrastive perplexity is much more stable than the original perplexity metric. Would the authors provide more detailed explanation or mathematical proof on the reasons for this? It is a bit unclear for me to understand this difference.

4. In the benchmark comparison, it would be better to include the GPT-3.5-Turbo-0613 and LongChat-13B-16k for comparison. Although they might be slower and more expensive, it would be helpful to understand that the cost from the compression.

**Questions:**

Please see the weakness.

---

> ### Author Response · Authors · 2023-11-21
> **Response to Reviewer PhaL**
>
> 1. **"the methods contains several aspects... it lacks a detailed ablation study that the influence... from each aspect..."**
> Thanks for the helpful suggestions. We have incorporated your suggestions and added detailed ablation study on different datasets in the updated version (Multi-document QA: Table 3; LongBench: Table 6; MuSiQue: Table 5). In general, removing any module in LongLLMLingua such as question-aware coarse-to-fine compression, dynamic compression ratio, document reordering, and subsequence recovery yield performance improvement for LLMLingua across different benchmarks.
>
> 2. **"For the reordering... whether the reordering will disturb the time-line between each document... whether this operation will disturb or introduce other difficulties for understanding?"**
> Thank you for pointing out the potential confusion. Since LLMs tend to be “lost-in-the-middle" as shown in Figure 1b, reordering can help LLMs better capture the most important information and thus benefit the overall performance. In fact, our empirical experiments reveal that LLMs have the ability to understand texts that are organized in an unnatural order. For example, in the timeline reorder task from LooGLE (Table 8), LongLLMLingua outperforms the original prompt by 15.9 points.
>
> 3. **"Figure3b, ...contrastive perplexity... Would the authors provide more detailed explanation or mathematical proof on the reasons for this?"**
> Thank you for the valuable feedback. We added a theoretical derivation of contrastive perplexity in Appendix B. The conclusion is that contrastive perplexity is positively correlated with $p(x^{\text{que}}|x_i, x_{<i})$, signifying the contribution of each token to question $x^{\text{que}}$. The larger the contrastive perplexity, the higher the $p(x^{\text{que}}|x_i, x_{<i})$, the greater the contribution of the token $x_i$, the higher its importance, and the more it should be retained. In addition, we included Figure 5 in Appendix D.1 to illustrate the distribution of document-level contrastive perplexity when the ground-true was placed at different positions (i.e., 1st, 5th, and 15th). It shows that even when key information is not at the beginning of the prompt, using contrastive perplexity can better capture token-level information related to the question.
>
> 4. **"In the benchmark comparison, it would be better to include the GPT-3.5-Turbo-0613 and LongChat-13B-16k for comparison."**
> Thanks for your suggestion. We have added experimental results using GPT-3.5-Turbo-0613(Table 2) and LongChat-13b-16k(Appendix D.5 Table 7) on LongBench. It demonstrates that the compressed prompts achieve higher performance than the original prompts by 4.8 points.

---

> > ### Comment · Reviewer_PhaL · 2023-11-23
> > **Reply to the Authors**
> >
> > Thanks for your detailed response. I have no further concerns and will keep my rate.

---

### Official Review · Reviewer_iM9S · 2023-11-03

**Soundness:** 2 fair
**Presentation:** 3 good
**Contribution:** 3 good
**Rating:** 5
**Confidence:** 4

**Summary:**

This paper builds upon the LLMLingua method, which utilizes a smaller LM to assess the perplexity of each token in a prompt, removing less informative tokens to compress the total length and reduce required compute. The authors identify several shortcomings of the method when applied to long inputs. Specifically, they conduct the perplexity evaluation conditioned on the question, implement a two-step compression with a dynamic compression threshold to first eliminate redundant documents and then compress the remaining context, and rearrange the remaining document to counteract the LMs' tendency to overlook crucial information when it appears in the middle of the context. From their evaluations, the authors ascertain that the LongLLMLingua method not only enhances the performance of black-box (API-based) LMs but also their accuracy.

**Strengths:**

The resulting technique is easy to use, and is general enough to be applicable in every long input scenario, including black-box api based LLMs. While several building blocks in the pipeline are specific to certain use-cases, the overall question-aware compression technique is applicable widely, and their results suggest it may be useful to simultaneously reduce costs and latency and even improve the downstream performance.

**Weaknesses:**

Some of the main contributions proposed in the paper are somewhat limited in their usefulness in a diverse set of naturally occurring tasks.
1. Removing documents and reordering is only relevant in the multi-document scenario where the relevant information is pertained in a small subset of the documents, and the rest are noise (e.g. in open-book question answering). Many use cases (e.g. long-input summarization and question answering, code understanding, multi-hop question answering etc.) will not benefit from it and may even be hurt by this procedure. Namely, in mult-hop question answering where one can determine the relevance of a downstream document only after answering the outer-part of the question, important documents may be filtered out with the procedure proposed in the paper.
2. Subsequence recovery is only relevant in very specific cases. Namely, where the task is extractive question answering. However, many real-world use cases that contain naturally occurring tasks over long inputs are not extractive rather generative.
3. The question-aware compression is only relevant when a question (or more generally, a long and specific instruction) is given. In use cases such as summarization, or conditional generation this is not the case.

All together, these specifications limit the contribution of the method when contrasting with the existing LLMLingua method which includes the remaining parts of the pipeline.

One of the main pieces of evidence in the paper is the evaluation on ZeroSCROLLS. However, I find this evaluation unsatisfactory, as the paper mentions they use the evaluation set of the benchmark, while Shaham et al. 2023 mentions explicitly that the evaluation set contains ``a small number of examples (~20 per task) in a "validation" split, meant for eyeballing purposes’’ and does not have enough statistical power to be used as statistical measurement (https://github.com/tau-nlp/zero_scrolls). Additionally, the authors do not provide a breakdown of the results to show that their method is indeed beneficial across the different scenarios.

The paper also considers several changes as significant contributions, but there are no ablation studies to show their usefulness. Namely, in §2 they mention the usage of $x^{doc} \| x^{que} \| x^{restrict}$ to compute the relevance of each document as an important contribution. An ablation on the ordering of the question and document should be added, as well as for the relevance of the restricting prompt. Moreover, all ablation studies that were performed were done on the NaturalQuestions dataset which is explicitly tailored for the proposed contribution. Specifically, one document contains the relevant information while the others are distractors. I would like to see the ablation study conducted on multi-document scenarios where this is not the case, such as as the multi-hop question answering scenario (e.g. MuSiQue [Trivedi et al., 2021] which also appears in ZeroSCROLLS).


The authors mention using a small LM to perform all pre-computation and compressions needed. However, they use LLaMA-2-7B-chat as their small LM, which has a significant overhead in itself, and may not be widely applicable to many use-cases. An ablation study on the performance of using smaller models should be added.

Comments on presentation and typos for the authors (not a weakness, but should be addressed prior to final publication):
1. In Page 8 (§5), the indexing of the two tables is reversed. Namely, Table 3 appears before table 2.
2. “Less cost” is used twice instead of “lower cost” (abstract and the main results paragraph in §5).
3. In the abstract, one of the sentences starts with “. and experimental …” where it should have been “. Experimental …”
4. “Derive costs” is used several times (including in the abstract) instead of “Drive costs”.
5. In §2 “ground-true” instead of “ground-truth”.
6. The styling guide indicates that third level headings (namely paragraph titles) should be in small caps, and not capitalized as appearing in the paper. See https://iclr.cc/Conferences/2023/CallForPapers for style information.
7. In §4.3 you denote the number of documents with $N_d$ while it was already denoted as $K$ beforehand.
8. In §4.3, equations (5) are a bit confusing, as multiple $x_i$ exist depending on $x_k$, thus multiple $\tau_i$ exist. This phrasing should be made more explicit for easier reading.

**Questions:**

1. In §3 you mention that for the distribution alignment, an instruction tuning of the small LM is performed. Was this finetuning done separately for each test case? If so, in the NaturalQuestions case, were the answer positions considered a specific case? Please provide more details on the finetuning procedure.
2. In page 5 §4.1 you say that “the tokens with high contrastive perplexities concentrate more on the left side of the dashed line, which corresponds to the document that contains the answer to question”. If I understand it correctly, this experiment was only done in the case where the ground-truth document is in the first position? If so, isn’t it  possible that the fact the contrastive learning approaches zero as the token position increases is simply a symptom of the “short-term memory” of perplexity (Khandelwal et al., 2018; Sun et al., 2021, Press et al., 2021a,b)?
3. Can you please provide a breakdown on the ZeroSCROLLS test set benchmark? Additionally, it would be helpful to compare your results with the baseline results of the same models where there were no token constraints.

**Details Of Ethics Concerns:**

No ethics considerations in the paper.

---

> ### Author Response · Authors · 2023-11-21
> **Response to Reviewer iM9S (1/2)**
>
> 1. **"Removing documents and reordering is only relevant in the multi-document scenario... Many use cases (e.g. long-input summarization, code understanding, multi-hop question answering etc.) will not benefit from it and may even be hurt by this procedure."**
> Thank you for pointing out the potential confusion. LongLLMLingua is designed as a general coarse-to-fine compression framework that allows us to adjust the granularity of compression based on the task at hand. When dealing with tasks such as long-input summarization where important information is evenly distributed throughout the original text, we can perform coarse-grained compression (i.e., “removing” in the question here) at the paragraph or sentence level rather than document level to improve the key information density. Since LLMs tend to be “lost-in-the-middle", reordering can help LLMs better capture the most important information and thus benefit the overall performance. We believe that the experimental results in long-input summarization (Table 6, LongBench summ.), single-/multi-doc question answering (Table 6, LongBench; Table 8, LooGLE), code understanding (Table 6, LongBench Code), multi-hop question answering (Table 5, MuSiQue), and Timeline reorder (Table 8, LooGLE) well demonstrate the effectiveness of LongLLMLingua. We have clarified this in the revised version.
>
> 2. **"Subsequence recovery is only relevant in... extractive question answering. However, many real-world use cases... are not extractive rather generative."**
> Yes, the benefit of subsequence recovery in certain generative scenarios is a little bit limited than it in extractive tasks. However, in most cases, the generated content may refer a lot to texts in the prompt in terms of entities or phrases, and subsequence recovery can effectively mitigate the information loss here. We conduct ablation study of subsequence recovery on diverse tasks for your information. As shown in Table 6, removing subsequence recovery would lead to a performance drop in Long Bench tasks “SingleDoc (QA)”, “muiltiDoc (QA)”, “Summ. (summarization)”, and “FewShot”.
>
> 3. **"question-aware compression is only relevant when a question (or more generally, a long and specific instruction) is given. In summarization, or conditional generation this is not the case."**
> The question-aware module, both coarse-grained and fine-grained, is designed for aware the changing of information density within the prompt. While it is crucial for tasks that involve following specific instructions (e.g., removing question-awareness would lead to a performance drop up to 35.1 points on NaturalQuestions as in Table 3), it can also be beneficial for summarization tasks and other tasks including multi-hop QA task. For example, Table 6 shows that in the LongBench summarization task, dropping question-awareness does lead to performance drop ~1-2points. And Table 5 shows that in the MuSiQue multi-hop QA, dropping question-awareness does lead to performance drop ~9 points. This suggests that question-awareness such as “please summarize this article” also provide valuable global context for the SLM, both coarse-grained and fine-grained, and have an implicit influence on the perplexity distribution during compression.
>
> 4. **"the above specifications limit the contribution of the method when contrasting with the existing LLMLingua method"**
> We hope the above responses have already addressed your concerns. The large, consistent performance gain from LLMLingua to LongLLMLingua across various tasks (as shown in Table 1-5 and 7, 8) also well demonstrates the contribution of LongLLMLingua.
>
> 5. **"the evaluation on ZeroSCROLLS is unsatisfactory, ...the evaluation set contains a small number of examples (~20 per task)... does not have enough statistical power…not provide a breakdown of the results to show that their method is indeed beneficial across the different scenarios."**
> Thanks for the suggestions. We use the validate set of ZeroSCROLLS for evaluation because the ground-true test set is inaccessible. We've added the breakdown results of ZeroSCROLLS in Appendix D.2 (Table 4). We further evaluate LongLLMLingua on MuSiQue (multi-hop QA, ~4.8k examples, Table 5) and LooGLE (long dependency QA, ~1.1k examples, Table 8). Experimental results show that compared with the original prompts, LongLLMlingua compressed prompts achieve comparable results on ZeroSCROLLS and outperform the original prompts by a large margin on MuSiQue and LooGLE, well validating its effectiveness in a wide range of scenarios.

---

> ### Author Response · Authors · 2023-11-21
> **Response to Reviewer iM9S (2/2)**
>
> 6. **"The paper also considers several changes as significant contributions, but there are no ablation studies to show their usefulness... An ablation on the ordering of the question and document should be added, as well as for the relevance of the restricting prompt. Moreover, all ablation studies that were performed were done on the NaturalQuestions... I would like to see the ablation study conducted on multi-document scenarios... such as the multi-hop question answering scenario (e.g. MuSiQue [Trivedi et al., 2021]..."**
> Thank you for your valuable suggestions. We have incorporated your suggestions and included an ablation study of reordering and restrict prompt on different datasets in the updated version (Multi-document QA: Table 3; LongBench: Table 6; MuSiQue: Table 5). In general, both reordering and the restrict prompt yield improvements for LongLLMLingua across different benchmarks.
>
> 7. **"The authors... use LLaMA-2-7B-chat as their small LM... An ablation study on the performance of using smaller models should be added."**
> Thanks for your feedback. we've added results with GPT2-small (using GPT2-dolly as the small LM) to Table 3, Table 5, and Table 6. The results show that GPT2-small compressed prompts can derive comparable even higher performance than the original prompts.
>
>
> 8. **"Typos"**
> Thanks for your thorough review. We have fixed these issues in the updated version.
>
> 9. **"...for the distribution alignment... Was this finetuning done separately for each test case? If so, in the NaturalQuestions case, were the answer positions considered a specific case? Please provide more details on the finetuning procedure."**
> Thanks for bringing up the misunderstanding. As mentioned in the section of "Implement details", for all experiments, we use llama2-7b-chat as the aligned version without any further fine-tuning.
>
> 10. **"In page 5 §4.1... this experiment was only done in the case where the ground-truth document is in the first position? If so, isn’t it possible that the fact the contrastive learning approaches zero as the token position increases is simply a symptom of the “short-term memory” of perplexity...?"**
> Thank you for the valuable comments. We would like to clarify that the distribution of contrastive perplexity is not caused by the diminishing effect. We have added a theoretical derivation of contrastive perplexity in Appendix B. The conclusion is that contrastive perplexity is positively correlated with $p(x^{\text{que}}|x_i, x_{<i})$, signifying the contribution of each token to question $x^{\text{que}}$. The larger the contrastive perplexity, the higher the $p(x^{\text{que}}|x_i, x_{<i})$, the greater the contribution of the token $x_i$, the higher its importance, and the more it should be retained. In addition, we include Figure 5 in Appendix D.1 to illustrate the distribution of document-level contrastive perplexity when the ground-true was placed at different positions (i.e., 1st, 5th, and 15th). It shows that even when key information is not at the beginning of the prompt, using contrastive perplexity can better capture token-level information related to the question.
>
> 11. **"Can you please provide a breakdown on the ZeroSCROLLS test set benchmark?... would be helpful to compare your results with the baseline results..."**
> We have added a breakdown of ZeroSCROLLS with the baseline results in Appendix D.2. In general, compared with the original results, LongLLMLingua achieving slightly higher numbers at 3x compression and maintaining the same level of performance at 6x compression.

---

### Official Review · Reviewer_qC5x · 2023-11-09

**Soundness:** 3 good
**Presentation:** 3 good
**Contribution:** 3 good
**Rating:** 6
**Confidence:** 4

**Summary:**

This paper proposes LongLLMLingua, a question-aware coarse-to-fine compression method to compress prompts and improve the key information density. The empirical results demonstrate that LongLLMLingua can substantially compress the prompts while maintaining the model performance and improving efficiency.

**Strengths:**

1. The method is novel. This paper proposes to discard irrelevant documents and tokens iteratively in a coarse-to-fine manner, which is novel and effective.
2. Measuring the token importance by perplexity is intuitive and insightful. Such an approach can be applied to black-box models, which is an advantage.
3. The empirical results are sound, demonstrating the effectiveness and efficiency of LongLLMLingua.

**Weaknesses:**

1. Prompt compression can be effective for tasks like QA, where the key information is sparsely distributed. In contrast to tasks, like summarization, the key information can be evenly distributed within inputs. Coarsely dropping a large amount of input may hurt the performance.
2. The inference latency can be improved by LongLLMLingua. However, the models need to evaluate the perplexity and compress prompts every time, which leads to non-trivial latency.

**Questions:**

1. How is the thresholding of $s_i$ determined to discard tokens of lower importance? Is it determined by the current budget $\tau_k$ of the documents?
2. Section 4.3 introduces a dynamic budget scheduler. It is unclear to me how the iteration is defined here. Is there an iterative evaluation of token importance?
3. I am interested in the perplexity distribution within sentences.  If the perplexity varies dramatically across tokens of one sentence, it can happen that a few tokens are retained sparsely, making the sentence inconsistent.

---

> ### Author Response · Authors · 2023-11-21
> **Response to Reviewer qC5x**
>
> 1. **"...tasks, like summarization, the key information can be evenly distributed within inputs. Coarsely dropping a large amount of input may hurt the performance."**
> Thank you for bringing up this point. LongLLMLingua is designed as a coarse-to-fine compression framework that allows us to adjust the granularity of compression based on the task at hand. For instance, when dealing with tasks where important information is evenly distributed throughout the original text, we can perform coarse-grained compression at the paragraph or sentence level. Additionally, the framework also allows us to adjust the proportion of content to be dropped during coarse-grained and fine-grained compression. For example, in summarization tasks, we can assign a smaller ratio to coarse-grained compression and a higher ratio to fine-grained compression to avoid dropping too much input during coarse-grained compression. We believe that our experiments in Table 2 (“summ”., i.e., summarization) provide evidence for the effectiveness of LongLLMLingua on tasks such as **summarization**. We have clarified this in the revised version.
>
> 2. **" The inference latency can be improved by LongLLMLingua. The models need to evaluate the perplexity and compress prompts every time, which leads to non-trivial latency."**
> Thank you for pointing out the potential confusion. Though LongLLMLingua evaluates perplexities every time which may cause some latency, the compression is done by a much smaller language model Llama-7B. Comparing to the latency caused by LLM inferencing on the compressed prompt, LongLLMLingua can still speedup the end-to-end system latency from 1.4x as shown in Table 1 to 2.3x as shown in Table 2 (Figure 5 in previous version).
>
> 3. **" How is the thresholding of $s_i$ determined to discard tokens of lower importance? Is it determined by the current budget $\tau_k$ of the documents?"**
> Yes, we discard tokens of lower importance, i.e., with $s_i$ smaller than the threshold, and the threshold is determined by the current budget $\tau_k$ of the documents.
>
> 4. **"...Is there an iterative evaluation of token importance?"**
> Thanks for the comments. We evaluate token importance in segment-level by iteratively conditioned on all previous compressed segments and calculate document-level threshold based on Eq. 5. We will add clarification in the final version.
>
> 5. **"If the perplexity varies dramatically across tokens of one sentence... making the sentence inconsistent"**
> Thanks for the question. Token perplexities do vary a lot within one sentence. Though removing tokens can render the sentence inconsistent and unreadable to human, it doesn’t appear to affect the ability of language models to comprehend the prompt. This phenomenon has been studied in Figure 4 of LLMLingua paper[1], which demonstrates that GPT-4 is capable of recovering the original details from the compressed, non-human-friendly prompt.
>
> [1] LLMLingua: Compressing Prompts for Accelerated Inference of Large Language Models. https://arxiv.org/abs/2310.05736

---

### Author Response · Authors · 2023-11-21
**General Response**

We greatly appreciate the comprehensive reviews and insightful feedback provided by each reviewer. In response to your feedback, we've made the following updates to our paper draft:

- We have included **theoretical derivations for contrastive perplexity** and a visual analysis of **document-level perplexity distribution** w.r.t. different ground-truth positions to illustrate the principles.
- We have incorporated extra experiments for the Multi-hop QA task, **MuSiQue**, and the long dependency benchmark, **LooGLE**, to demonstrate the effectiveness of LongLLMLingua in tasks requiring global information, such as multi-hop QA and timeline reorder.
- We have expanded our paper with the **breakdown results of ZeroSCROLLS** and more **detailed ablation experiments** on NaturalQuestions, LongBench, and MuSiQue to demonstrate the contribution of each component in LongLLMLingua in a wide range of tasks.
- We have added experiments on **LongBench using the LongChat-13b-16k** as the LLMs for compression.
- We have included end-to-end **latency** evaluation for various methods on NaturalQuestions, LongBench, and ZeroScrolls.

We thank the reviewers again for their thorough evaluations and look forward to your further feedback. Please do not hesitate to reach out if you have any additional questions or require further clarification.

---

### Meta-Review · Area_Chair_wr1G · 2023-12-21

**Metareview:**

This proposes LongLLMLingua, which is based on the LLMLingua method. These methods utilizes a smaller LM to assess the perplexity of each token in a prompt, removing less informative tokens to compress the total length and reduce required compute. The authors identify several shortcomings of the method when applied to long inputs. Specifically, they conduct the perplexity evaluation conditioned on the question, implement a two-step compression with a dynamic compression threshold to first eliminate redundant documents and then compress the remaining context, and rearrange the remaining document to counteract the LMs' tendency to overlook crucial information when it appears in the middle of the context. From their evaluations, the authors ascertain that the LongLLMLingua method not only enhances the latency/cost of black-box LMs but also their accuracy. This paper evaluates on long context tasks such as single-/multi-document QA, few-shot learning, summarization, synthetic tasks, and code completion rather than short GSM8K, BBH, ShareGPT, and Arxiv-March23 in the original LLMLingua.

Reviewers generally liked the effectiveness of the method as shown in the experiments presented. They like how the method is generally applicable in black-box models. Reviewers also liked parts of the method such as coarse-to-fine (qC5x) and question aware compression (iM9S). The biggest drawback is that reviewers were not convinced by the improvements over the base work of LLMLingua and the quality of ablations to show the effectiveness of the few key improvements described in section 4.

Reviewers qdfA and iM9S both think the few additional points of improvements are not significant enough over LLMLingua. The AC considers the contrastive question aware method to be potentially significant, but tend to agree with the reviewer that the re-ordering to avoid loss in the middle and subsequence recovery are indeed simple adaptations to the new tasks. Though the reverse conditioning and contrastive scoring has appeared in previous work and applies naturally here as well.

The authors gave detailed responses to most points including two new evaluations and new ablation experiments showing that question-aware coarse grain and perhaps the reverse probability to contribute most significantly to the performance. Most reviewers did not get a chance to re-evaluate the paper in-light of these significant new results. Though the ablation do seem to confirm the initial suspicion of the reviewer that the source of improvements were opaque in the initial paper.

In the end, given unenthusiastic reviews, shortcomings in the original paper, and the overall method of using queries to Llama-7B to get better prompt for GPT, the AC recommends reject. To accept this paper, at least some reviewers should be enthusiastic about the empirical strength of the results and the paper should clearly identify the source of improvement.

**Justification For Why Not Higher Score:**

reviewers identified critical issues. Against the reviewer majority since the negative reviewer identifies clear issues whereas the slightly positive reviews were not nearly as informed.

**Justification For Why Not Lower Score:**

N/A

---

### Decision · Program_Chairs · 2024-01-16

Reject